# Regulation of Flavonoid Biosynthesis by the MYB-bHLH-WDR (MBW) Complex in Plants and Its Specific Features in Cereals

**DOI:** 10.3390/ijms26020734

**Published:** 2025-01-16

**Authors:** Andrey N. Bulanov, Elena A. Andreeva, Natalia V. Tsvetkova, Pavel A. Zykin

**Affiliations:** 1Department of Genetics and Biotechnology, Saint Petersburg State University, 7/9 Universitetskaya Embankment, 199034 Saint Petersburg, Russia; an.bulanov20002014@gmail.com (A.N.B.); n.tswetkowa@spbu.ru (N.V.T.); 2Laboratory of Plant Genetics and Biotechnology, N. I. Vavilov Institute of General Genetics, Russian Academy of Sciences, 119333 Moscow, Russia; 3Department of Cytology and Histology, Saint Petersburg State University, 7/9 Universitetskaya Embankment, 199034 Saint Petersburg, Russia; pavel.zykin@spbu.ru

**Keywords:** transcription factors, MYC-like bHLH, R2R3-MYB, WDR, flavonoids, maize, rice, barley, wheat

## Abstract

Flavonoids are a large group of secondary metabolites, which are responsible for pigmentation, signaling, protection from unfavorable environmental conditions, and other important functions, as well as providing numerous benefits for human health. Various stages of flavonoid biosynthesis are subject to complex regulation by three groups of transcription regulators—MYC-like bHLH, R2R3-MYB and WDR which form the MBW regulatory complex. We attempt to cover the main aspects of this intriguing regulatory system in plants, as well as to summarize information on their distinctive features in cereals. Published data revealed the following perspectives for further research: (1) In cereals, a large number of paralogs of MYC and MYB transcription factors are present, and their diversification has led to spatial and biochemical specialization, providing an opportunity to fine-tune the distribution and composition of flavonoid compounds; (2) Regulatory systems formed by MBW proteins in cereals possess distinctive features that are not yet fully understood and require further investigation; (3) Non-classical MB-EMSY-like complexes, WDR-independent MB complexes, and solely acting R2R3-MYB transcription factors are of particular interest for studying unique regulatory mechanisms in plants. More comprehensive understanding of flavonoid biosynthesis regulation will allow us to develop cereal varieties with the required flavonoid content and spatial distribution.

## 1. Introduction

Plants have a wide variety of pigments that can give their tissues and organs almost any color. Flavonoid compounds are some of the most important plant pigments, along with chlorophylls, carotenoids, and betalains. The primary flavonoid pigments in plants are anthocyanins, while in cereals, proanthocyanidins and phlobaphenes also play an important role as colorants. Flavonoids perform many functions necessary for the plant: serve as phytoalexins, toxins against insects and herbivorous animals [1,2,3], protect the plant from excessive light [4,5,6,7], function as pigments [8,9,10] or signaling molecules [11]. Flavonoids are also widely used by humans, which makes research in this field of particular interest.

The pathway of flavonoid biosynthesis is well studied, its various stages are subject to complex regulation, which determines the temporal, spatial, and biochemical specificity of accumulation of these substances. A key role is played by the MYC-like bHLH (avian Myelocytomatosis virus-like basic Helix-Loop-Helix) and R2R3-MYB (Repeat 2—Repeat 3 avian Myeloblastosis virus) transcription factors (TF), and WDR (WD repeat or WD40 repeat) proteins, which form the so-called MBW protein complex that directly regulates the expression of structural genes [12,13,14]. Members of bHLH, MYB and WDR protein families are present in almost all groups of eukaryotes and widely represented in plants.

In plants, the first cloned genes of MYC and MYB proteins were *Lc* (*Leaf color*) and *C1* (*Colorless 1*) from maize, respectively [15,16]. The transcription factors C1 and Lc were among the first studied in plants, which, together with the WDR coregulators, activate the anthocyanin biosynthesis. They form the MBW complex which recognizes regulatory elements in the promoters of structural biosynthetic genes and activates their expression. This is the one of the best-studied plant regulatory systems [12,17,18,19,20]. The formation of such a complex is a distinctive feature of the regulation of flavonoid biosynthesis, as well as of the formation of epidermal structures, such as trichomes and root hairs [17,18,19,20]. MYC and MYB transcription factors determine the temporal, spatial, and biochemical specificity of these processes and are responsible for the recognition and binding of specific DNA sequences [12,14,21,22,23,24,25], while WDR proteins stabilize the MYC-MYB complex, thereby enhancing its activity [12,19,26,27,28].

The accumulation of flavonoid compounds in cereals is a very important agricultural trait, since it strongly affects plant vitality and fertility [29,30], as well as the quality of produced foods due to their distinctive taste, bright color and impact on animal and human health [31,32,33,34,35,36,37]. Breeding of cereals such as maize, rice, barley, wheat and rye has been mainly aimed at reducing the content of flavonoid compounds in their grains, especially anthocyanins, to maximize calorie content and have white flour, therefore the majority of modern cultivated varieties are characterized by the absence of these secondary metabolites, or by their reduced amount, not only in the grain but also in the vegetative organs of the plant [38,39,40,41,42]. However, because of presumed beneficial effects on human health [32,33,34], attempts have been made to develop cultivated varieties enriched with flavonoid compounds.

Manipulations with the genes of MYC and MYB regulators make it possible to effectively introduce the trait into cultivated varieties [43,44,45,46]. Though the main features of MBW complex formation and flavonoid regulation in cereals are shared with dicotyledonous plants, some unique traits were described. One of them is a large number of paralogs of the MYC and MYB transcription factors regulating flavonoid biosynthesis in cereals of which *R* locus in maize is a bright example. Numerous alleles of *R* locus are known, which may cause a lack of anthocyanin pigmentation, have pigmentation only in vegetative organs, only in generative organs, or in both. A competitive MBW complex formation, which is described in dicotyledons, which is involved in the regulation of flavonoid biosynthesis, is not known for cereals. However, in rice unique hierarchical regulatory systems including MYC, MYB, and WDR proteins are described. Another important aspect is the potential presence of multiple WDR coregulators of flavonoid biosynthesis in cereals or the independence of MB complexes from them, in contrast to dicotyledons. It can be assumed that, at least in some cereals, the regulatory system formed by MYC and MYB transcription factors may function independently of WDR proteins [16,47,48,49,50,51,52,53,54,55,56,57].

Manipulations with the genes of MYC and MYB regulators make it possible to effectively introduce the trait into cultivated varieties [43,44,45,46]. A large number of paralogs of the MYC and MYB transcription factors regulating flavonoid biosynthesis in cereals, as well as the complex regulation of their activity and their incompletely studied interactions, allow them to have extremely diverse phenotypes due to the accumulation of anthocyanins and other flavonoid compounds [16,47,48,49,50,51,52,53,54,55,56,57].

The above makes the studies of flavonoid biosynthesis regulation in cereal plants not only quite interesting from the point of view of fundamental science but also has great practical significance. In this review, we try to generalize the accumulated knowledge about the features of the structure and functioning of MBW proteins and their mutual regulation in plants, in particular, in the model object *Arabidopsis thaliana* L., maize *Zea mays* L. and some other species, as well as their similarities and differences in dicotyledonous and monocotyledonous plants. The second part systematizes rather scattered information about these regulators and their genes in cereal plants, as well as their distinctive features, intricacies of their functions and interactions with each other.

The articles included in the review were selected based on their relevance to the main scope of the manuscript and their contribution to the current understanding of the regulation of flavonoid biosynthesis in cereals and other plants by MBW complexes. For articles directly addressing MYC-like bHLH, R2R3-MYB, and WDR proteins and their genes regulating flavonoid biosynthesis, original research articles were prioritized to ensure that the cited works were primary sources of discovery. The main focus is on studies of maize, rice, barley, and wheat, as these are the most widely used and, consequently, studied cereal crops. For the general aspects of flavonoid biosynthesis regulation in plants, articles focused on *A. thaliana*, petunia (*Petunia hybrida* Vilm.), snapdragon (*Antirrhinum majus* L.), and maize (*Z. mays*) were prioritized, as these are the best-studied model systems in the field. Research providing novel insights or significantly broadening knowledge in the field was included. However, less-cited or niche articles were also incorporated if they presented unique or essential findings.

## 2. Flavonoid Compounds and Flavonoid Pigments

Flavonoids, which include such important pigments, as the anthocyanins, proanthocyanidins, and phlobaphenes, are a large group of phenolic compounds, important secondary metabolites of plants with extremely diverse functions. Since the first description of the chemical properties of these plant pigments in 1664 by Robert Boyle, flavonoids were used in some of the greatest scientific discoveries. Among them is Gregor Mendel’s description of the inheritance of pea (*Lathyrus oleraceus* Lam., also known as *Pisum sativum* L.) traits, one of which was the seed color, determined by the content of anthocyanins, as well as Barbara McClintock’s discovery of transposon elements affecting the genes of flavonoid biosynthesis in maize [10]. Recently, special attention has been paid to the study of this group of secondary metabolites not only due to their important functions in plant organisms, but also because of their high biological activity, impact on food quality, human and animal health [31,32,33,34,35,36,37].

Flavonoids are characterized by the presence of two aromatic rings connected by a three-carbon chain with a backbone described as C6–C3–C6. The mentioned three-carbon chain can form a pyran ring containing an oxygen atom. Flavonoids can be subject to aromatic ring substitutions at different positions by –OH, –OCH3 or –CH3 groups, as well as glycosylation by various mono- and disaccharides and their modifications with the addition of various hydroxycinnamic (caffeic, ferulic, *p*-coumaric) acids, malonic acid and other residues [58]. The three-carbon chain of the molecule can be in different oxidation states, depending on which the following main groups of flavonoids are distinguished [10]: chalcones, aurones, flavanones, flavan-4-ols, phlobaphenes, flavones, flavonols, isoflavonoids, dihydroflavonols, leucoanthocyanidins (flavan-3,4-diols), flavan-3-ols, proanthocyanidins (condensed tannins), and anthocyanidins, which glycosides are called anthocyanins (Figure 1).

Proanthocyanidins, phlobaphenes, and anthocyanins are the primary flavonoid pigments in plants, including cereals [10,59]. Proanthocyanidins and phlobaphenes are synthesized by polymerisation of flavan-3-ols and flavan-4-ols, respectively. Proanthocyanidins are accumulated in the various vegetative and generative organs of many plants, including Arabidopsis seed coat [60]. Many forms of barley, wheat, and rye also contain proanthocyanidins in the seed coat (testa) [42,61]. Their oxidized forms determine the characteristic color of red-grained wheat varieties [62]. The red coloration of rice is also mainly due to the accumulation of proanthocyanidins in the pericarp [63]. The red coloration of maize grains, unlike other cereals, is due to the accumulation of phlobaphenes, in addition to anthocyanins [64]. Phlobaphenes are found in only a few plant species, including maize and sorghum (*Sorghum bicolor* L. Moench) [10].

Anthocyanidins and anthocyanins contain a trivalent oxygen atom, oxonium, in the pyran ring [8,65,66]. They are accumulated in the plant mainly in the form of glycosides in vacuoles, which is facilitated by the enzyme glutathione S-transferase (GST) [67,68,69]. Due to the different patterns of aromatic rings substitutions, the compounds of this group can have an extremely diverse color, which varies from pink or blue to red or purple. The shade of the color of plant tissue or organ depends, among other things, on the pattern of glycosylation, covalent bonding with other flavonoids, the pH of the environment and the formation of non-covalent bond with metal cations [8,70]. Due to the above-described properties, anthocyanins, along with carotenoids, are the main non-photosynthetic pigments of many plants (with the notable exception of plants from the order Caryophyllales, the most important pigments of which are betalains [9]), their main function is the pigmentation of certain parts of the body to attract pollinators or seed dispersers. Various combinations of accumulated anthocyanin and carotenoid pigments determine the wide range of plant petals and fruits shades [8,9]. In maize, anthocyanins accumulate primarily in the aleurone, as well as in the pericarp, and are mainly represented by cyanidin derivatives [38,63,71]. Black rice varieties are characterized by a high content of cyanidin-3-glucoside and peonidin in the pericarp, as well as delphinidin derivatives in the aleurone layer [38,63]. Purple pigmentation of the pericarp of the grain in barley, wheat, and rye is associated with the accumulation of cyanidin-3-glucoside and peonidin, their acylated derivatives, cyanidin-3-rutinoside, and some others [42,72]. In addition to the purple color, the grains of these species may also have a blue or gray-green coloration, associated with the accumulation of the blue anthocyanin delphinidin-3-rutinoside and others in the aleurone layer. The combination of these two traits leads to the black color of caryopsis [38,42,63,73,74,75].

The chemical nature of flavonoids allows them to protect the green organs of plants from excessive light. For example, under stress conditions (damage by pathogenic organisms, salinization, deficiency of macro- and micro-nutrients, hypothermia or moisture deficiency) the plant’s ability to utilize the products of light reactions of photosynthesis is greatly reduced, which can lead to excessive accumulation of substances with a strong oxidative or reducing potential [76]. To prevent the damage, in response to stress factors many plant organs accumulate flavonoid compounds [77,78,79,80,81,82]. Flavonoid compounds are also involved in signaling between plants and bacteria or fungi. Thus, flavonoids are released by plants during the formation of mycorrhiza [83,84,85,86,87,88] or bacterial nodules [89,90] and affect the vital activity of the corresponding symbionts. Some flavonoids are plant-produced antibiotics [91,92,93], others prevent the plant from being eaten [94,95] or are involved in the response to strong competition with other plants [96]. In some plants flavonoids are shown to have an important effect on male fertility. In tomato pollen grains normally contain flavonols, which are thought to prevent the formation of reactive oxygen species. Mutant tomato plants that are unable to accumulate flavonols have defective pollen with damaged cell walls [29]. In contrast, in rice, various classes of flavonoid compounds accumulate in pollen grains, so mutants with reduced amounts of flavonols do not completely lose fertility, probably due to similar effects of other classes of flavonoids [30,97].

Many flavonoid compounds are potent antioxidants and may have beneficial effects on human health [32,33,34,37]. Anthocyanins also could have beneficial effects on human health in other aspects [31,35,36,98,99,100,101,102,103,104,105]. However, the bioavailability of anthocyanins is extremely low compared to other flavonoids [106]. Anthocyanins can be used as natural food colorings, but it is associated with significant difficulties due to their instability, however there are a number of methods that nevertheless allow the use of anthocyanins as a worthy and useful replacement for synthetic dyes [107].

## 3. Flavonoid Biosynthesis Pathway

All phenylpropanoid compounds of plants, including flavonoids, are synthesized from phenylalanine or tyrosine (Figure 1). The first and key reaction of the phenylpropanoid biosynthesis pathway for these secondary metabolites is the deamination of phenylalanine, catalyzed by the enzyme phenylalanine ammonia lyase (PAL) or phenylalanine/tyrosine ammonia-lyase (PTAL), which is found in grasses [108]. Further transformations are catalyzed by the enzymes cinnamic acid 4-hydroxylase (C4H), hydroxycinnamate-CoA ligase (4CL) and chalcone synthase (CHS), which catalyzes the formation of chalcones—the simplest representatives of flavonoids. All groups of flavonoid compounds are the downstream products of chalcone [10,109]. Subsequent sequential interconversions of flavonoids are catalyzed by the enzymes chalcone isomerase (CHI), flavone-3, -3′ and -3′5′-hydroxylases (F3H, F3′H, F3′5′H), and dihydroflavonol-4-reductase (DFR), which lead to the formation of flavan-3,4-diols. Leucoanthocyanidin reductase (LAR) and anthocyanidin synthase (ANS) catalyze the conversion of flavan-3,4-diols into flavan-3-ols, which are polymerized into proanthocyanidins, and into anthocyanidins, respectively. Flavan-3-ols can also be synthesized from anthocyanidins by the action of anthocyanidin reductase (ANR) [10,109]. Normally, anthocyanidins do not accumulate in plant cells, but undergo further modifications: glycosylation (enzymes—UDP-glucose: flavonoid-3-glycosyltransferase, UFGT; anthocyanin-5-O-glycosyltransferase, 5GT), rhamnosylation (UDP-rhamnose: anthocyanidin-3-glycoside rhamnosyltransferase, 3RT), acylation (anthocyanin 3- or 5-aromatic acyltransferase, 3AT or 5AT), methylation (anthocyanin 3′- or 5′-O-methyltransferase, A3′OMT or A3′5′OMT) and many others (Figure 1) [8,59,109].

Structural genes (encoding the necessary enzymes) of anthocyanin and proanthocyanidin biosynthesis are conventionally divided into early genes: PAL, C4H, 4CL, CHS, CHI, F3H, F3′H, F3′5′H, and late genes: DFR, ANS, LAR, ANR, and anthocyanidin modification enzymes [10,109]. The initial experimental models of flavonoid biosynthesis pathway genetic control were maize (*Zea mays* L.), snapdragon (*A. majus*) and petunia (*P. hybrida*) [10]. The chalcone synthase gene from parsley (*Petroselinum hortense* Hoffm.) was the first isolated structural gene of flavonoid biosynthesis [110] and served as a basis for the discovery of its homologs in other plants. Twelve chalcone synthase genes have been found in petunia, four of which are expressed (CHSA, CHSB, CHSD and CHSJ) [111]; two genes (C2 and WHP1) have been identified in maize; and one gene (Nivea) has been found in snapdragon [109]. Subsequently, many structural biosynthetic genes were characterized in these plants, and for some of them spatial variations in expression were shown. One of the petunia chalcone isomerase genes, CHIA, is expressed in all flower tissues, as well as in seedlings exposed to UV radiation, while another gene, CHIB, is expressed exclusively in immature anthers [112].

## 4. General Mechanisms of Regulation of Flavonoid Biosynthesis by MBW Complexes

### 4.1. Features of MYB, bHLH, and WDR Transcription Factors Regulating Anthocyanin Biosynthesis

#### 4.1.1. R2R3-MYB Transcription Factors

MYB and bHLH are among the largest families of plant transcription factors [113]. The first cloned genes of the MYB family were the v-myb (viral-myb) gene of avian myeloblastosis virus and its cellular homolog *c-myb* (*cellular-myb*) [114]. Later, such genes were found in almost all groups of eukaryotic organisms, including all groups of plants [115]. The first cloned gene of a plant transcription factor was the *C1* gene of maize, recessive alleles of which lead to the absence of anthocyanins in the embryonic organs of the grain. *C1* was shown to encode a transcription factor with 40% similarity to animal *c-myb* type proto-oncogenes, providing the first evidence of the involvement of these TFs in the regulation of anthocyanin biosynthesis [16].

MYB transcription factors contain DNA-binding domains of the helix-turn-helix type, present in the molecule in the form of one or more imperfect repeats of 51–52 amino acid residues, designated R (Repeat). Each such R domain contains three highly conserved tryptophan residues (tryptophan cluster). However, only the second repeat (R2) in such a hydrophobic core has a cavity, which plays a key role in DNA binding and the ability for trans-activation of genes [116,117]. Transcription factors of the MYB family common to plants (MYB3R) and animals (c-MYB) belong to the R1R2R3 type and have a fairly similar structure (3 unequal R-repeats) and, probably, the function related to the control of cell proliferation. Most plant MYB proteins belong to a specific type—R2R3-MYB transcription factors that have lost the first R-repeat. They are involved in the regulation of secondary metabolism, hormonal response, and cell type specification [115,118]. Diverse members of the R2R3-MYB family have been identified as key regulators of flavonoid biosynthesis, providing positive regulation of structural genes [18,19,20,21,22,27,28,47,64]. Proteins of the plant R3-MYB (1R-MYB) family, which have lost the R1- and R2-repeats, negatively regulate flavonoid biosynthesis or other processes regulated by MBW complexes by binding to bHLH proteins and interfering with R2R3-MYB transcription factors, thus providing additional regulation [20,27]. R2R3-MYB transcription factors that regulate flavonoid biosynthesis bind to the promoters of structural and regulatory genes through two types of nucleotide motifs—MBS, MYB binding site (CNGTTR), or AC-rich, including ACI ([A/C]CCAAC[C/G]), ACII (ACCAACC), ACIII (ACCTA[A/C]), and ARE, anthocyanin regulatory element (A[C/A]C[T/A]AC) [119,120,121,122,123].

The family of R2R3-MYB transcription factors is one of the largest families in plants, with hundreds of highly diverse members [18,124,125]. One of the first classifications of *Arabidopsis* MYB proteins, based on phylogenetic analysis of their amino acid sequences, divided them into 22 subgroups. For each subgroup, a specific amino acid sequence motifs were identified that lie outside the R2R3 domains and indicate a specific function [124]. Further study of the family allowed the number of its subgroups to be increased to 26, and the specific motifs for each of them were revised [18]. Proteins regulating anthocyanin biosynthesis belong to phylogenetically close subgroups: subgroup 5, including *Arabidopsis* TT2 (TRANSPARENT TESTA 2) and maize C1 and Pl (*Purple leaf*) proteins; subgroup 6, including *Arabidopsis* PAP1, PAP2 (Production of anthocyanin pigment) and petunia AN2 (Anthocyanin 2) proteins; and subgroup 7, including maize P1 (Pericarp color 1) protein. Similar to these proteins, *Arabidopsis* MYB factors GL1 (GLABRA 1) and WER (WEREWOLF), regulating the formation of epidermal structures, belong to subgroup 15, phylogenetically close to subgroups 5–7 [18,124].

Interestingly, no R2R3-MYB proteins of subgroup 6, which are present in all other angiosperms, have been found in cereal plants. All their known MBW complexes are formed by R2R3-MYB proteins of subgroup 5, which is characterized by the presence of a conserved DExWLRx2T motif in the C-terminal region of the protein [18,20,126]. However, the *Kala3* (*Key gene for black coloration by anthocyanin accumulation on chromosome 3*) gene product of rice, an R2R3-MYB protein belonging to subgroup 5 with a conserved motif [V/L][W/I]x2KAxRCT, also has a motif [K/R]P[Q/R]P[Q/R][S/T]F, which is specific to R2R3-MYB transcription factors of subgroup 6 [18,127]. Such a “chimeric” structure, combining the features of subgroups 5 and 6, is not typical of the known R2R3-MYB regulators of anthocyanin pigmentation of other cereal plants and dicots, but is found in R2R3-MYB TFs of orchids (Orchidaceae) [127].

For R2R3-MYB proteins of subgroups 5, 6 and 15 the formation of complexes with subgroup IIIf bHLH transcription factors has been shown. For such binding, the presence of a conservative motif [D/E]Lx2[R/K]x3Lx6Lx3R in the consensus sequence of the R3 repeat is necessary [20]. On the other hand, transcription factors ZmP1 and OsP1, belonging to subgroup 7, do not interact with bHLH proteins and independently regulate phlobaphene and anthocyanin pigmentation, respectively [46,64,124,126,128,129,130,131,132]. This inability to interact is associated with disruption of their bHLH-binding motif [20,126]. Therefore, classification of R2R3-MYB transcription factor to groups 5–7 (and possibly group 15) may indicate that it is involved in the regulation of anthocyanin pigmentation, and the presence of the above-described motif in the R3 repeat indicates that it forms complexes with bHLH proteins.

#### 4.1.2. MYC-like bHLH Transcription Factors

The MYC protein gene was first discovered in the genome of the avian myelocytomatosis virus and was designated *v-myc* (*viral-myc*), and its cellular homolog-oncogene was designated *c-myc* (*cellular-myc*) [133]. The *Lc* gene of maize, recessive alleles of which lead to the absence of anthocyanin pigmentation of leaves and other plant organs, was the first described plant gene of MYC-like transcription factor [15]. Subsequently, genes encoding proteins with a highly conserved bHLH (basic-Helix-Loop-Helix) domain, which is a key feature of these TFs, were found in all groups of plants, green and red algae. A common origin with the corresponding animal genes was also shown. Each plant genome contains a large number of bHLH family genes with extremely diverse functions [17,134,135].

The key role in the DNA-binding activity of bHLH proteins is played by the bHLH domain, which contains a basic region and a region with a helix-turn-helix structure [136]. A common property of transcription factors of the bHLH family is the binding of a specific DNA sequence—the E-box, carrying the CANNTG motif. The basic region of the bHLH domain, which contains a highly conserved HER motif (His^5^-Glu^9^-Arg^13^), is responsible for this binding [17,137,138]. Anthocyanin-related MYC-like bHLH transcription factors specifically bind to the E/G box (CANNTG/CACGTG) or AN1-like motifs (CAGATG and CATCTG) [119,139,140]. In addition to the DNA-binding activity, the bHLH domain is also responsible for the dimerization of the proteins carrying it. Homo- or heterodimerization of bHLH proteins is carried out by the α-helices of the HLH region, with the key role played by Ile, Leu and Val, located at similar positions in plant and animal proteins [17,137,138].

The bHLH family of transcription factors in plants includes up to hundreds of members in each species. Phylogenetic analysis of *Arabidopsis* bHLH transcription factors identified 12 subgroups within this family [141]. Further study of hundreds of bHLH proteins allowed the expansion of this classification to 26 subgroups [17]. All of them have a conserved bHLH domain structure. The classification is based on the presence of special sequences in representatives of each subgroup that belong to certain functional domains, the exon-intron structure of their genes, and phylogenetic similarity. It was shown that most of the known bHLH proteins involved in the regulation of anthocyanin and proanthocyanidin biosynthesis belong to subgroup IIIf and have a similar structure. These include, for instance, the transcription factors GL3 (GLABRA 3), ELG3 (ENHANCER OF GLABRA3), and TT8 (TRANSPARENT TESTA 8) of *Arabidopsis*; R (Red), B (Booster), and Lc of maize; OsB1, OsB2 (*Oryza sativa* Booster 1, 2), Rb, and Rc (Red b and c) of rice; AN1 (Anthocyanin 1) of petunia [17,141]; and GL3L and TT8L of freesia (*Freesia* × *hybrida* L.H.Bailey) [142]. All of them form complexes with R2R3-MYB proteins to activate structural genes for anthocyanin and/or proanthocyanidin biosynthesis [12,132,142,143,144]. A common property of subgroup IIIf bHLH transcription factors is the presence of a MYB-binding region (MIR, MYB-interacting region) in the *N*-terminal part of the protein. They have several conserved amino acid sequences within the subgroup, which belong to the MIR region [17,141,142]. Another common property of bHLH proteins of subgroup IIIf is their binding to WDR transcription factors, whereas proteins of phylogenetically close subgroups IIId and IIIe are not capable of forming complexes with WDR [20,28].

However, the bHLH transcription factor gene of another subgroup, IIIb, the product of *PpbHLH46* gene of red pear (*Pyrus persica* Pers.), recently shown to be capable of activating anthocyanin biosynthesis. The gene-encoded protein, like the subgroup IIIf proteins, was shown to form a complex with the MYB transcription factor, as well as to be able to bind to the WDR factor. Such a complex is capable of binding to the *PpUFGT* promoter and activating its expression. Apparently, bHLH transcription factors of subgroup IIIb can also form MBW complexes [145]. The classification of bHLH transcription factors to subgroups IIIf or IIIb may indicate their involvement in the regulation of structural anthocyanin biosynthesis genes expression and the formation of MBW complexes.

It is important to note that, since the discovery of bHLH proteins regulating anthocyanin biosynthesis was associated with the observation of shared homology with the mammalian Myc oncogene [15], such proteins are often referred to as MYC-like or simply MYC rather than bHLH in a number of species [15,55,57,146,147,148,149,150,151], and particularly in cereals [55,146,152]. Since then, the term ’MYC-like bHLH’ has been exclusively used to refer to subgroup IIIb [153], III(d+e) [147,151,154], and IIIf [57] proteins [17,141,147], as they share the common characteristic of possessing an acidic *N*-terminal region [148,150,153,155], as do the maize proteins of the R/B family [126,156,157]. Thus, in this paper, the term ‘MYC-like bHLH’ or ‘MYC proteins’ refers to anthocyanin-related subgroup IIIf, as well as subgroup IIIb and III(d+e) bHLH proteins.

#### 4.1.3. WDR Proteins

For the functioning of described *Arabidopsis* MYB and bHLH transcription factors the presence of a third protein, TF encoded by the *TTG1* gene (*TRANSPARENT TESTA GLABRA 1*), is required [13,14,158,159]. Mutations in this gene result in both the absence of trichomes and the absence of anthocyanins in the plant. *TTG1* encodes a WDR family protein, a widespread group of regulatory proteins in eukaryotes [160]. WDR proteins (WD-Repeat, another name is WD-40) contain conservative repeats of 23–41 amino acids, each of them is almost always preceded by a pair of amino acids—glycine (G) and histidine (H). The next amino acid residues in the polypeptide chain after the repeat are almost always tryptophan (W) and aspartic acid (D), which gave the name to the entire family [161]. Such repeats were first discovered in a β-subunit of the human G protein [162]. In plants, the first WDR protein gene discovered was the *AN11* (*Anthocyanin 11*) gene of petunia, which controls the anthocyanin coloration of the flower. The gene encodes a protein that contains five WD repeats 26–28 amino acid residues long, located between the GH and WD sequences [163]. *Arabidopsis* TTG1 has a high amino acid sequence similarity with AN11 [160]. A homologous gene for the WDR regulatory protein was later identified in maize. Mutations of *PAC1* (*Pale Aleurone Color 1*) gene result in the absence of anthocyanins in grain tissues [164]. The protein encoded by this gene has a high amino acid sequence similarity with TTG1 and AN11 [26].

Thus, regulation of the expression of structural anthocyanin biosynthesis genes requires simultaneous presence of functional alleles of the genes encoding proteins of three families: R2R3-MYB, MYC-like bHLH and WDR.

### 4.2. Structure of MBW Complexes

The presence of an active transcription factor of only R2R3-MYB or only MYC-like bHLH type in a plant is not sufficient to activate anthocyanin biosynthesis; the expression of structural genes is triggered by the simultaneous presence of TFs of both families. The first evidence for this was obtained in the study of genes of the *R*/*B* family and the *C1* gene of maize. *C1* determines the anthocyanin pigmentation of embryonic tissues in grain and encodes the R2R3-MYB transcription factor [16]. The *Pl* gene is highly homologous to *C1* and determines the color of vegetative plant organs [47]. Different alleles of the *R* and *B* loci also determine the anthocyanin pigmentation of many plant organs [165]. Coding sequences of the alleles of these loci, *R-S*, *B-Peru* and *B-I*, are highly homologous to another member of the R family mentioned above—the *Lc* gene [15] and encode MYC transcription factors [166,167]. It was also found that for the full functioning of the *R* or *B* gene, the presence of a functional allele *C1* is necessary [168,169], and a little later, a direct interaction of the B and C1 proteins was proven [156].

Genes of *Arabidopsis* homologous to the *ZmR*/*B* and *ZmC1*/*Pl* genes have both similar (regulation of anthocyanin and/or proanthocyanidin biosynthesis) and different functions (regulation of trichome formation). The *GL* gene determines trichome formation and encodes the R2R3-MYB transcription factor [170]. *GL3* and *EGL3* encode MYC TF and are responsible for the formation of epidermal structures [13,14]. The *PAP1*, *PAP2* and *TT2* genes encode MYB transcription factors that activate the expression of structural genes for anthocyanin and/or proanthocyanidin biosynthesis [159,171]. The *TT8* gene is also responsible for the proanthocyanidin biosynthesis and encodes the MYC-like TF [158]. As for the B and C1 proteins in maize, the formation of complexes activating the expression of various genes was shown for the MYB and MYC pairs of *Arabidopsis* transcription factors: GL1 and GL3 [13], EGL3 and GL1, EGL3 and PAP1 [14], EGL3 and TT2, TT2 and TT8 [20]. MYB-bHLH complexes have also been described in vertebrates; for example, they are formed by the human transcription factors c-MYB and MyoD [172].

The first experimental evidence of a protein complex that simultaneously includes all three types of these transcription factors was obtained in a study of the GL3, GL1 and TTG1 proteins of *Arabidopsis* [13]. The authors showed that GL3 is capable of binding to both GL1 and TTG1, while GL1 is incapable of binding to TTG1. However, the possibility of heterodimerization of GL3 with some other MYC factor was not excluded [13]. Such a possibility of different MYC proteins to bind to each other was demonstrated [19], however, the functional role of such interactions is unclear. Direct evidence for the existence of a ternary complex was obtained a bit later [12]. Studying the activation of the *Arabidopsis* leucoanthocyanidin reductase *BAN* (*BANYULS*) gene promoter by the transcription factors TT2, TT8 and TTG1, they found that the full expression of this gene was possible only with the simultaneous presence of all three proteins. The TT2-TT8 complex is capable of activating gene expression independently, but its level positively correlates with the expression of TTG1. The TT8, like GL3, binds to both TT2 and TTG1. At the same time, the possibility of simultaneous binding of all three molecules was proven. Such a ternary complex is capable of directly binding to the *BAN* gene promoter. TT8 could be replaced by another homologous MYC transcription factor, but the less the homology between such a MYC protein and TT8, the less the activation of gene expression. TT2, on other hand, cannot be replaced by any other MYB transcription factor for successful gene expression [12]. These suggested that MYB proteins are responsible for the specific binding of structural gene promoters in ternary complexes, while MYC factors provide acceptable conformational states of their molecules for contact with DNA [12]. However, there are also examples of complexes in which the DNA-binding activity of MYC transcription factors plays an important role (see Figure 2 and Figure 3B,C,E,F) [119,126,139,140,173].

Another interesting feature of the interaction between MYC and MYB proteins was revealed in the study of maize transcription factors. In the R family of TFs, an ACT-like (aspartate kinase, chorismate mutase, and tyrA) domain was found in the C-terminal part of the protein, in addition to the bHLH domain. This domain is necessary and sufficient for the homodimerization of the molecule [126,157]. It has homology with the ACT domain of phosphoglycerate dehydrogenase of *Escherichia coli* (PGDH). Its secondary structure is similar to that of PGDH and also promotes homodimerization. In contrast, elements located in the *N*-terminal part of the molecule are responsible for the binding of R proteins to R2R3-MYB transcription factors [126,157]. Dimerization via the ACT domain is also shown for GL3, although with lower affinity than with R [174].

The protein of another family, RIF1 (R-interacting factor 1), was found to be able to bind to R [175]. RIF1 has conserved sequences corresponding to the ENT domain of chromatin modification factors, which classifies it as a family of EMSY-like transcription factors. The protein also contains sequences of the AGENET domain of the Royal Family of proteins, but neither ENT nor AGENET are able to bind to R alone, so it is assumed that multiple structures of the protein determine the interaction with MYC proteins [175]. In the absence of this protein, plants with normal expression of the *R* and *C1* genes exhibit less anthocyanin pigmentation [139]. The action of the R-C1 complex was found to be associated with acetylation of histone 3 (H3K9/14) in the *A1* gene promoter, suggesting the involvement of R via RIF1 in chromatin remodulation (Figure 3B) [175].

RIF1 itself is not able to bind the dihydroflavonol reductase *A1* gene promoter, but MYB regulator C1 recruits it to the regulatory region of the gene. In this case, dimerization of R proteins via bHLH domains prevents their interaction with RIF1, i.e., RIF1 interacts only with R monomers and is recruited to the promoters of structural genes via an alternative complex with C1 [139]. Thus, a model was proposed describing the ACT domain of R as a switch between two alternative protein conformations and, accordingly, the protein complexes formed by them (Figure 3B) [139]. According to this model, in one state, monomeric R proteins, each bound to a single RIF1 molecule, are recruited via C1 to the promoter of an early structural gene, *A1*, in particular. These proteins dimerize through ACT domains, binding two RIF1 molecules to histones, which in turn attract chromatin remodeling factors and activate gene expression. In another state, ACT domains are inactive, and R molecules dimerize via their bHLH domains, which enables the MB/MBW complex to directly bind via R to the G-box of the promoter of another structural gene, *Bz1* (*Bronze-1*) encoding anthocyanidin 3-*O*-glucosyltransferase (Figure 3B) [139]. A later study showed that R, in the absence of C1, is only capable of directly binding to the *Bz1* gene promoter, but is unable to bind to the *A1* promoter without a MYB partner, which partially confirmed the model described above [176]. These data are consistent with the fact that the MYB protein P1, which is incapable of forming MB complexes [20,126,157] and activates the expression of early flavonoid biosynthetic genes independently (Figure 3A [64,128,129,130]) is incapable of activating the expression from the *Bz1* promoter due to the requirement for the presence of MYC protein binding sites for this [121,139,177]. The ability to dimerize via the ACT domain was also demonstrated for the *Arabidopsis* MYC transcription factor GL3, but with much lower affinity, which is associated with the presence of a polar amino acid (S595) at a certain position in its ACT domain, in contrast to R, which contains an aliphatic residue (V568) at the corresponding position. The S595V and V568S substitutions lead to an increase in the efficiency of GL3 ACT domains homodimerization and a disruption of R ACT domains homodimerization, respectively; however, the presence of Val at this position does not correlate with the possibility of such dimerization in other MYC proteins [174]. Nevertheless, the presence of ACT-like domains in many MYC-like bHLH proteins, including those regulating flavonoid biosynthesis [143,174], suggests that such alternative regulation via the ACT domain may also be characteristic of other transcription factors, at least evolutionarily close to R. Thus, at least in cereal plants, alternative regulation of anthocyanin biosynthesis by MB/MBW complexes via the ACT domains of MYC proteins is possible. It is not known whether WDR proteins are involved in this process; however, it can be assumed that PAC1 or similar proteins stabilize the MB complex formed by the dimerization of the bHLH domains of MYC transcription factors [12,13,19,26], in contrast to RIF1, which stabilizes an alternative complex with dimerization of MYC proteins through their ACT domains [139,143,175] (Figure 2j,k and Figure 3B). Possible variants of regulation of flavonoid biosynthesis gene expression by MYC-like bHLH, R2R3-MYB, and WDR transcription factors are summarized in Figure 2.

**Figure 2 ijms-26-00734-f002:**
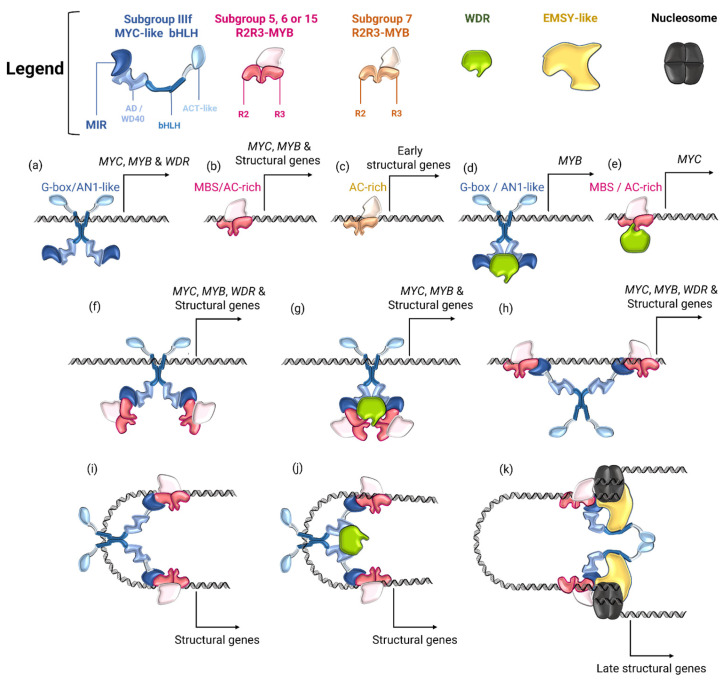
Variants of positive regulation of flavonoid biosynthesis gene expression by MYC-like bHLH and R2R3-MYB transcription factors and WDR regulatory proteins. The MYC-like bHLH, MYB, WDR, EMSY-like proteins, and the nucleosome, as well as their functional domains, are shown in different colors, as indicated in the legend. (**a**–**e**) MYC-like bHLH proteins bind to promoters via the E/G box or AN1-like motifs [119,139,140], MYB proteins bind via the MBS (MYB-binding site) or AC-rich motifs [119,120,121,122,123]. (**a**)—activation/repression of MYC-like bHLH, MYB and WDR coregulator genes expression by MYC protein dimers [44]; (**b**)—activation/repression of MYC and MYB coregulator genes expression and structural genes by MYB proteins of subgroups 5, 6 and 15 [157]; (**c**)—activation of early structural genes expression by subgroup 7 MYB proteins [129,132]; (**d**)—activation of MYB coregulator expression by the BW complex [27,178]; (**e**)—activation of MYB coregulator expression by the MW complex [179]; (**f**–**h**)—activation/repression of MYC, MYB and WDR coregulator genes expression, as well as structural genes by the MB or MBW complexes via the DNA-binding activity of MYC (**f**,**g**) or MYB (**h**) transcription factors; (**i**,**j**)—activation of structural gene expression by MB or MBW complexes, respectively, through the DNA-binding activity of both MYC and MYB transcription factors [19,44,173,175,178,179]; (**k**)—activation of late biosynthetic gene expression by the alternative EMSY-like-MB complex formed during homodimerization of MYC proteins through the ACT domain with recruitment of the EMSY-like factor to nucleosomes [139,175].

### 4.3. Regulatory Networks of MBW Complexes

For *Arabidopsis*, cotton (*Gossypium hirsutum* L.), the Alpine rock-cress (*Arabis alpina* L.), and petunia (*P. hybrida*), it was shown that some MYC-like bHLH proteins compete with MYB for binding to WDR (the process of the competitive complex formation) (Figure 3C) [27,28]. In this case, the presence of certain MYB factors increases the binding of the MYC factor to WDR (in particular, AtPAP1 and AtPAP2). In other combinations of MYC and MYB, the formation of the complex is disrupted. For a number of other MYC factors, competition with MYB for binding to WDR was not detected. Phylogenetic analysis of the MYC transcription factors used in the study showed that they fall into two clades. Almost all TFs of the first clade (for example, GhDEL65, AtGL3 and AtEGL3 and their homologs AaGL3 and AaEGL3) exhibit such competitive binding, while proteins of the second group (such as AtTT8, AaTT8, PhAN1, ZmLc and ZmR-S) do not exhibit it (Figure 3C,D) [28]. Interestingly, replacing just one amino acid in the protein AtGL3 of the clade I (its normal function is to form trichomes and root hairs [13]) with another amino acid common in this position to clade II proteins (e.g., AtTT8, which causes accumulation of flavonoid compounds [158]) results in a dramatic change in the phenotype, as well as a lack of competition between MYB and WDR for binding to AtGL3 [28]. Thus, *Arabidopsis* plants of the *gl3egl3tt8* phenotype do not synthesize flavonoids and do not form trichomes [14]. In transgenic plants of this genotype, with overexpression of the normal *AtGL3* gene, flavonoids are synthesized and trichomes are formed on the leaves. In plants with overexpression of the mutant *AtGL3*, with the F177I substitution, only anthocyanin pigmentation is restored, but trichomes are absent [28]. Competitive complex formation in *Arabidopsis* seems to be the key feature of trichome formation regulation (Figure 3B), and, apparently, does not play a significant role in the regulation of flavonoid biosynthesis (Figure 3C). Thus, to bind to promoters of genes regulating flavonoid biosynthesis, it requires the presence of AtGL1 or AtTTG1 [27,144,180]. At the same time, the recruitment of AtGL3 to the promoters of the R3-MYB repressor *AtCPC* (*CAPRICE*) gene and the key trichome differentiation specifier *AtGL2* apparently requires the presence of the AtGL1 protein exclusively and is impaired when the double complex is exposed to the WDR factor AtTTG1. On the other hand, when AtGL3 forms a complex with AtTTG1 it is capable of activating the promoter of another R3-MYB gene, *AtTRY* (*TRYPTICHON*). This activity is also disrupted by the presence of the AtGL1 protein (Figure 3C) [27]. Interestingly, normal AtGL3 is able to bind to the promoter of its own *AtGL3* gene independently of its MYB partner AtGL1 and, together with R3-MYB repressors, reduce the level of its expression [27,144,180]. This is assumed to be the key process in cell differentiation into a trichome cell. In a cell where AtGL1 activity prevails, the expression of the *AtGL2* gene is activated, initiating the subsequent trichome formation cascade. In contrast, in a cell where AtTTG1 activity prevails, the AtTRY and AtCPC suppress *AtGL3* expression, preventing such differentiation. (Figure 3C) [27,180].

Unlike in *Arabidopsis*, in snapdragon (*A. majus*) and petunia (*P. hybrida*) the effect of competitive complex formation on flavonoid biosynthesis was shown. The *A. majus* clade I MYC protein AmDel (Delila) forms a complex with the R2R3-MYB protein AmROS1 (Rosea1) and, most likely without the involvement of the WDR protein, triggers the expression of anthocyanin biosynthesis genes. The clade II MYC protein AmInc1 (Incolorata1), on the other hand, forms a complex with AmROS1, which is most active in the presence of the AmWDR1 protein, as part of the MBW complex. In this case, the competitive complex formation creates a hierarchical system: Del-ROS1 MB complex activates the expression of *AmInc1* and *AmWDR1*, the products of which form the ROS1-Inc1-WDR1 MBW complex, which triggers anthocyanin biosynthesis (Figure 3E(a)). The authors suggest that the presence of an active AmWDR1 protein, at the same time, reduces the activity of Del-ROS1 MB complex, thus forming a potential negative feedback loop [173]. A similar hierarchical system was found in petunia, in which the R2R3-MYB repressor PhMYB27 competes with the WDR protein PhAN11 for binding to the clade I MYC protein PhJAF13. The reduction of *PhMYB27* gene expression allows PhAN11 to bind to PhJAF13 (Figure 3E(b,c)), consequently activating anthocyanin biosynthesis through the activation of the MBW complex formed by the clade II MYC protein PhAN1, WDR protein PhAN11, and various R2R3-MYB activator proteins (Figure 3E(e,f)). This system also involves a third step: the activation of R3-MYB repressors of anthocyanin biosynthesis, which forms a negative feedback loop (Figure 3E(g)); however, it is unknown whether it is universal for all plants [178].

Similar relationships have been described in *Arabidopsis*, where the clade I MYC proteins AtGL3 and AtEGL3 activate the expression of the clade II protein AtTT8. However, this is not the key process in *Arabidopsis*, since AtTT8 is able to autonomously activate its own expression (Figure 3F(a,b)) [12,14,21,27,179]. Thus, R2R3-MYB and WDR transcription factors compete for binding to clade I MYC proteins, such as AtGL3, while the formed alternative MYC-WDR (BW) or MYB-MYC (MB) complexes have different functional activities. The presence of clade I MYC proteins has not yet been shown in monocots, suggesting that competitive complex formation and the corresponding hierarchical regulatory systems may be traits specific to eudicots [27,28].

In kiwifruit (*Actinidia chinensis* Planch.) an alternative hierarchical regulatory system was found in which clade II MYC transcription factors, AcbHLH4 and AcbHLH5, form MBW complexes, which trigger the expression of both the *AcWDR1* gene, rapidly amplifying its amount, and a third clade II MYC gene, *AcbHLH1*, the product of which, in turn, triggers the expression of structural genes of anthocyanin biosynthesis (Figure 3E(h,i)) [119].

In addition to the hierarchical relationships between different MYC proteins and the MBW complexes they form, MYB-WDR complexes (MW) were also found to regulate the activity of their MYC partners (Figure 3F(a,b)). For instance, *Arabidopsis* MYB transcription factors AtTT2 or AtPAP1 together with the WDR factor AtTTG1 are capable of activating the expression of the *AtGL3*, *AtEGL3* and *AtTT8* genes [159,179]. Interestingly, AtTT8 itself is also capable of binding to the promoter of its own gene and inducing its expression [179]. This suggests that AtPAP1/TT8 and AtTTG1 trigger self-sustaining expression of the *AtTT8* gene to form the MBW complex, which triggers anthocyanin biosynthesis (Figure 3F(a,b)), thus synchronizing the *AtTT8* expression and the phenotype caused by it [179]. The same self-sustaining expression was found in *AcbHLH1*, the product of which is capable of activating its own expression, as well as the expression of its MYB partner gene *AcMYBF110* (Figure 3E(i)) [119].

Another unique hierarchical system of MBW gene regulation was discovered in rice (Figure 3F(c–f)). The MYC transcription factor OsB2 (other names are S1 and Kala4 [44,181]) is able to directly activate the expression of its MYB partner *OsC1* gene through the E-boxes in its promoter. The initiation of *OsC1* expression through the AC-rich element (CCTACC), in turn, activates the expression of the WDR gene *OsPAC1* (*OsTTG1*, *WA1*), the product of which forms an MBW complex with OsB2 and OsC1 triggering anthocyanin biosynthesis (Figure 3F(c–e)) [44]. However, with such cascade regulation, an excess of MBW complexes can quickly accumulate in cells, potentially disrupting regulatory processes. Therefore, the authors suggest the presence of a negative feedback loop that maintains the expression level of these genes at a constant, optimal level. This is evidenced by the fact that overexpression of *OsB2* (*Kala4*/*S1*), on the contrary, leads to a decrease in the number of *OsC1* gene transcripts (Figure 3F(f)) [44]. The regulatory networks described above, formed by MYC-like bHLH, R2R3-MYB and WDR proteins, are shown in Figure 3.

However, MBW complexes that directly bind to the promoters of anthocyanin biosynthesis genes and regulate their expression, are themselves finely regulated by hormones and hormone-dependent transcription factors, light- and temperature-dependent proteins, non-coding RNA, sugar signaling factors, MYB and bHLH repressors and many other factors [145,182,183,184,185,186,187,188,189,190,191,192,193,194,195,196,197,198,199,200]. Such regulation can occur at all levels of MBW protein functioning, ranging from chromatin remodeling at the loci encoding them or interactions with their gene transcripts, to covalent modifications of these proteins or direct protein-protein interactions with the entire MBW complex or its individual components, which has recently been comprehensively reviewed in a number of excellent studies [189,190,191,192,194,198].

In summary, the regulation of flavonoid biosynthesis is carried out by a complex interaction of three main groups of transcription factors: MYC-like bHLH, R2R3-MYB, and WDR. Their not yet fully understood complex interactions determine the temporal, spatial and biochemical specificity of the accumulation of flavonoids. At the same time, understanding which MYC and MYB transcription factors are expressed at a specific moment in time is crucial, as their interactions can lead to extremely diverse manifestations of traits.

**Figure 3 ijms-26-00734-f003:**
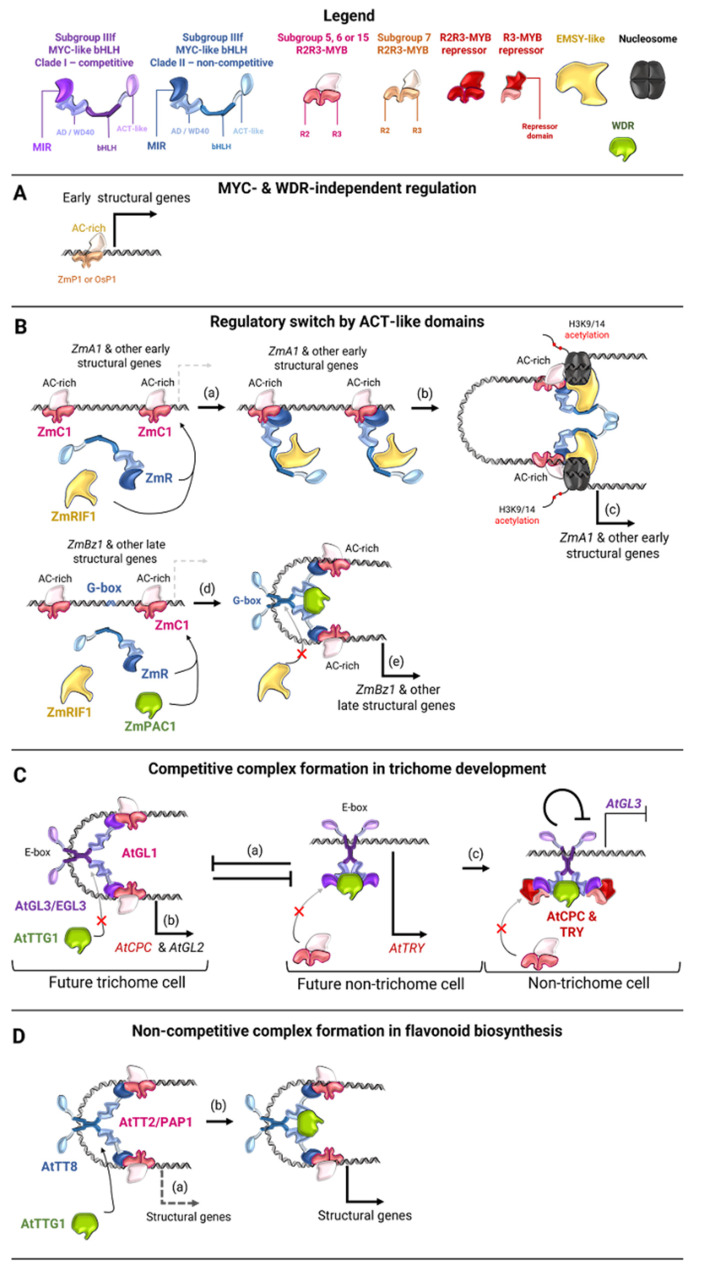
Variants of regulatory networks formed by MYC-like bHLH and R2R3-MYB transcription factors and WDR regulatory proteins. The subgroup IIIf MYC-like bHLH of clades I and II, R2R3-MYB positive and negative regulators, R3-MYB repressors, WDR and EMSY-like regulatory proteins, and the nucleosome, as well as their functional domains, are shown in different colors, as indicated in the legend. Red crosses indicate either the inability of proteins to bind to the complex or the absence of these proteins. (**A**) Subgroup 7 R2R3-MYB proteins of maize and rice are not capable of forming complexes with MYC proteins and trigger the expression of early structural biosynthetic genes alone [64,129,130]. (**B**) Model of regulatory switch by ACT-like domain involving EMSY-like coregulators [139]. (**a**) Regulatory elements in the promoters of the early flavonoid biosynthesis gene *ZmA1* are recognized by R2R3-MYB proteins ZmC1, which recruit MYC proteins ZmR and EMSY-like proteins ZmRIF1 to them. (**b**) In the absence of a ZmR binding site, this leads to dimerization of MYC proteins via their ACT-like domain and formation of a complex, (**c**) which allows ZmRIF1 protein, probably via binding to nucleosomes, to recruit chromatin remodeling factors that acetylate H3K9/14 and trigger expression of the structural gene. (**d**) ZmC1 proteins are recruited to the regulatory elements of the late biosynthetic gene *ZmBz1* promoter and are capable of activating its expression with very low efficiency. (**e**) The presence of a G-box in the *ZmBz1* promoter allows *ZmR* binding and dimerization through the bHLH domains, which triggers gene expression without the participation of ZmRIF1 [121,126,139,143,157,175,177]. The WDR protein ZmPAC1 is likely involved in this process [19,26]. (**C**) Regulation of trichome formation in *Arabidopsis* as example of competitive MBW-complex formation (**a**) Binding of the R2R3-MYB protein AtGL1 to the IIIf MYC-like bHLH protein of the clade I AtGL3 in *Arabidopsis* prevents the binding of the WDR coregulator AtTTG1 (formation of the BW complex), and vice versa [27,28]. (**b**) In the cell in which the activity of the AtGL3 protein predominates, the expression of the *AtGL2* gene is activated, which triggers a regulatory cascade, leading to its differentiation into a trichome. (**c**) In a cell with predominant AtTTG1 activity, the expression of *AtGL2* is not triggered, and the R3-MYB repressors AtTRY and AtCPC imported from the neighboring trichome cell suppress the activity of AtGL1, which prevents differentiation into a trichome [27,144,180]. (**D**) Regulation of flavonoid biosynthesis in *Arabidopsis* as an example of competitive MBW-complex formation. (**a**) In *Arabidopsis* subgroup IIIf MYC-like bHLH protein of the clade II AtTT8 forms MB-complex with R2R3-MYB proteins AtTT2 or AtPAP1, which is capable of binding to promoters of structural genes and triggering their expression with low efficiency. (**b**) The activation efficiency of structural gene expression increases significantly with the formation of MBW-complexes involving the WDR protein AtTTG1, which stabilizes them. At the same time, R2R3-MYB and WDR proteins do not compete for binding to the MYC protein [12,27,28,158,179]. (**E**) Hierarchical regulation among different subgroup IIIf MYC-like bHLH proteins in flavonoid biosynthesis. (**a**–**g**) Examples of hierarchical regulation between subgroup IIIf MYC-like bHLH proteins of the clades I and II. (**a**) In snapdragon, the MB complex of the clade I MYC protein AmDel and the R2R3-MYB protein AmROS1 is capable of triggering the expression of both *AmDFR* and, probably, other structural genes, as well as the genes for the clade II MYC protein AmInc1 and the WDR protein AmWDR1. At the same time, such activity is suppressed by the AmWDR1 protein, which competes for AmDel binding. The accumulation of AmInc1 and AmWDR1 proteins leads to the formation of the MBW complex, which highly efficiently triggers the expression of *AmDFR* and, probably, other structural genes [173]. (**b**) In petunia, under conditions where no accumulation of flavonoid compounds is required, the R2R3-MYB repressor gene *PhMYB27* is expressed and suppresses the expression of the R2R3-MYB biosynthesis activator genes. (**c**) The MB complex of PhMYB27 and clade I MYC protein PhJAF13 suppresses the expression of structural and regulatory genes of flavonoid biosynthesis. At the same time, the formation of such a complex prevents the binding of the WDR protein PhAN11 to PhJAF13 (**d**). When a signal for the need for flavonoid biosynthesis is received, the expression of *PhMYB27* is suppressed, and the expression of R2R3-MYB activators is triggered. (**e**) This allows the formation of the MW complex of the PhJAF13 and PhAN11 proteins, which triggers the expression of the gene for clade II MYC protein PhAN1. (**f**) The accumulation of PhAN1 leads to the formation of the MBW complex, which triggers the expression of structural genes. (**g**) This MBW complex also activates the expression of the R3-MYB repressor gene *PhMYBx*, which forms a negative feedback loop [28,178]. (**h**,**i**) Examples of hierarchical regulation between subgroup IIIf MYC-like bHLH protein of the clade II in kiwifruit. (**h**) MBW complexes of the clade II MYC proteins AcbHLH4 and AcbHLH5 with R2R3-MYB protein AcMYBF110 and WDR protein AcWDR1 specifically recognize AN1-like regulatory elements in the promoter of the gene for another clade II MYC protein, AcbHLH1, and trigger its expression. (**i**) The accumulation of AcbHLH1 leads to the formation of an MBW complex with AcMYBF110 and AcWDR1, which specifically recognizes the G-box in the promoters of structural biosynthetic genes. This MBW complex also activates the expression of *AcbHLH1* and *AcMYBF110*, which forms a positive feedback loop. (**F**) Hierarchical regulation between subgroup IIIf MYC-like bHLH, R2R3-MYB, and WDR proteins in regulation of flavonoid biosynthesis. (**a**,**b**) An example of *MYC* gene expression activation by the MW complex in *Arabidopsis*. (**a**) The MW complex of MYB protein AtTT2 and WDR protein AtTTG1 is capable of triggering the expression of the gene for their MYC partner AtTT8. Accumulation of AtTT8 leads to the formation of MBW complexes with AtTT2 or AtPAP1 and AtTTG1, triggering the expression of structural genes of biosynthesis. (**b**) Moreover, such a complex (**c**–**f**) is an example of a unique hierarchical regulatory system in rice [12,14,19,21,28,123,158,179]. (**c**) The MYC protein OsB2 can independently recognize the E-box in the promoter of its MYB partner gene *OsC1*. (**d**) Accumulation of OsC1 leads to the formation of an MB complex capable of recognizing AC-rich elements in the promoter of the WDR coregulator gene *OsPAC1*. (**e**) Accumulation of OsPAC1 leads to the formation of an MBW complex by these three proteins, which effectively activates the expression of structural biosynthetic genes. (**f**) It is assumed that such an MBW complex suppresses *OsB2* expression, forming a negative feedback loop [44].

### 4.4. Genes of MBW Regulators of Flavonoid Biosynthesis in Cereal Plants

#### 4.4.1. Diversity of MBW Genes in Maize (*Zea mays* L.)

In maize, there is a much wider diversity of MBW complex genes regulating anthocyanin pigmentation than in the dicot species mentioned above (Table 1, Table 2 and Table 3).

The large family of bHLH transcription factor genes *R*/*B* includes the complex *R* locus. Numerous alleles of *R* locus are known, which have extremely diverse phenotypes. Depending on the *R* allele a plant carries, it may lack anthocyanin pigmentation, have pigmentation only in vegetative organs, only in generative organs, or in both (Table 2) [165]. Such a large variation in phenotypes is associated with the structure of the locus: in certain alleles, two main components are distinguished, functional variants of which are present in the *R-r* allele (*Red*-*red*, red aleurone and red plant, respectively) [201,202]. The first component contains a single copy of the MYC-like bHLH transcription factor gene, *P* (*Plant*), and is responsible for the pigmentation of vegetative organs [50]. In contrast, the second component contains two tandemly duplicated sequences of the MYC protein genes (*S1* and *S2*, *Seed*) that share a promoter containing transposable element *doppia* responsible for the pigmentation of aleurone. A third component, q, contains a duplicated promoter region homologous to the promoter of the P gene [52,166,203]. The formation of such a complex structure was accompanied by the insertion and excision of transposons, which mediated duplication of *R* sequences [24,52,203]. In addition to the alleles of the locus containing the *P* or *S* genes, there are numerous alleles of this locus that have a different structure and cause other phenotypes, including anthocyanin coloration of almost all plant tissues and organs in various combinations [23,24,50,52,165,203,204,205,206]. A brief description of some of them can be found in Table 2 and Figure 4. Other loci of the *R* family genes that determine anthocyanin pigmentation have also been studied, each containing single-copy MYC transcription factor genes. These include *Lc*, which is responsible for the color of such maternal-origin organs as the leaf midrib, glumes, leaf auricles, and pericarp [15,207,208]; *Sn* (*Scutellar node color*), which regulates root and leaf pigmentation, and whose light-dependent expression is also observed in the mesocotyl, scutellum, and pericarp [24,209,210]; *Hopi*, which determines the pigmentation of the root, leaves, mesocotyl, anthers, pericarp, as well as light-induced pigmentation of the aleurone and scutellum at late stages of embryo development [23]; and *PSH* (*Purple leaf sheaths*), which is responsible for the formation of the purple color of the leaf sheath and also determines the color of the stem, leaf, and root exposed to light [211]. Interestingly, the *R* locus genes have a constitutive expression level in embryonic tissues, while *Sn* and *Hopi*, on the contrary, enhance their pigmentation in response to light at the early and late stages of caryopsis development, respectively [23,24]. The above-mentioned *R* family genes (*Sn*, *Hopi*, and *Lc*), including multiple MYC protein genes of the *R* locus, are located within a small region of chromosome 10 and have high nucleotide sequence similarity, indicating that they were formed as a result of multiple duplications (Figure 4) [15,23,24,50,52,201,202,203,204,205,206,207,209,210,212]. Such concentration of paralogs in one region of the chromosome leads to significant instability of this region and high mutability, primarily associated with unequal crossing-over events. This has led to the enormous diversity of the *R* locus alleles, which has been further enhanced by multiple chromosomal rearrangements mediated by transposon insertions [15,23,24,50,52,54,201,202,203,204,205,206,207,209,210,212]. In this light, *Hopi* can be considered as one of the alleles of the *R* locus, formed as a result of a recent chromosomal rearrangement [23]. As a result of these rearrangements, a large set of paralogous *R* family genes apparently emerged in maize, gradually specializing in certain functions, which is primarily associated with the specialization of their regulatory regions determining the spatial and, in part, temporal specificity of anthocyanin accumulation [15,23,24,25,48,50,52,54,156,201,202,203,204,205,206,207,209,210,212,213,214,215].

**Table 1 ijms-26-00734-t001:** Brief description of characterized loci, genes and proteins of MYC-like bHLH, R2R3-MYB, and WDR regulators of anthocyanin pigmentation in maize (*Zea mays* L.).

Family	Gene (Locus)	Chromosome	Function	MBW Complex Formation	Note	References
MYC-like bHLH	*ZmR* (*Red*)	10	Positive regulator of anthocyanin pigmentation of almost all tissues and organs, depending on the allele (see Table 2)	Yes, with ZmC1/Pl and ZmPAC1	A complex locus, different alleles contain varying numbers of copies of *MYC* genes and other sequences (see Table 2 and Figure 4)	[26,50,52,126,139,143,157,165,174,175,176,201,202,203,204,205,206,207,212,216]
	*ZmLc* (*Leaf color*)	10	Positive regulator of anthocyanin pigmentation of leaf midrib, glumes, leaf auricles and pericarp	Yes, with ZmC1/Pl and, probably, ZmPAC1	-	[15,207,208]
	*ZmSn* (*Scuttelar node color*)	10	Positive regulator of anthocyanin pigmentation of root, leaf, mesocotyl, scutellum and pericarp	Yes, with ZmC1/Pl and ZmPAC1	Expression is induced by light in the mesocotyl and also in the early stages of embryo development in the scutellum and pericarp	[24,209,210]
	*ZmHopi*	10	Positive regulator of anthocyanin pigmentation of roots, leaves, mesocotyl, anthers, pericarp, scutellum and aleurone	Yes, with ZmC1/Pl and ZmPAC1	Expression is induced by light in the late stages of embryo development in the scutellum and aleurone	[23]
	*ZmB* (*Booster*)	2	Positive regulator of anthocyanin pigmentation of vegetative tissues and pericarp, depending on the allele	Probably, with ZmC1/Pl and ZmPAC1	Many different alleles with different promoter structures (see Table 3 and Figure 5)	[15,23,24,25,28,47,52,121,126,156,167,169,175,210,217,218]
	*ZmPSH* (*Purple leaf sheath*)	10	Positive regulator of anthocyanin pigmentation of leaf sheath, stem, leaves and root	Unknown	Expression in the root is activated by light	[211]
	*ZmIN1* (*Intensifier 1*)	7	Negative regulator of anthocyanin pigmentation of aleurone	Probably, a competitor of C1/Pl–R/B–PAC1 complexes	IN1 likely competes with R family proteins for binding to MYB and WDR due to a defective structure	[26,219]
R2R2-MYB	*ZmC1* (*Colorless 1*)	9	Positive regulator of anthocyanin pigmentation of aleurone	Yes, with R/B family and ZmPAC1	-	[15,16,22,23,26,28,47,126,156,157,164,169,220,221]
	*ZmPl* (*Purple leaf*)	6	Positive regulator of anthocyanin pigmentation of vegetative tissues and pericarp	Yes, with R family and, probably, with ZmB and ZmPAC1	-	[19,23,24,47,121,126,156,164,167,169,175,201,217,218]
	*ZmP * (*Pericarp color*)	1	Positive regulator of phlobaphene biosynthesis in pericarp, panicle, cob, silk and anthers	No	Contains two (or more) highly homologous *P* genes that activate the expression of early genes of flavonoid biosynthesis and activate the biosynthesis of phlobaphenes independently of MYC proteins	[46,64,128,129,130,131,222]
WDR	*ZmPAC1* (*Pale* *aleurone color 1*)	5	Positive regulator of anthocyanin pigmentation of roots and aleurone	Yes, with R/B family and ZmC1	Expression is observed in many plant tissues, but mutations only result in the absence of root and aleurone pigmentation.	[19,26,164]

**Figure 4 ijms-26-00734-f004:**
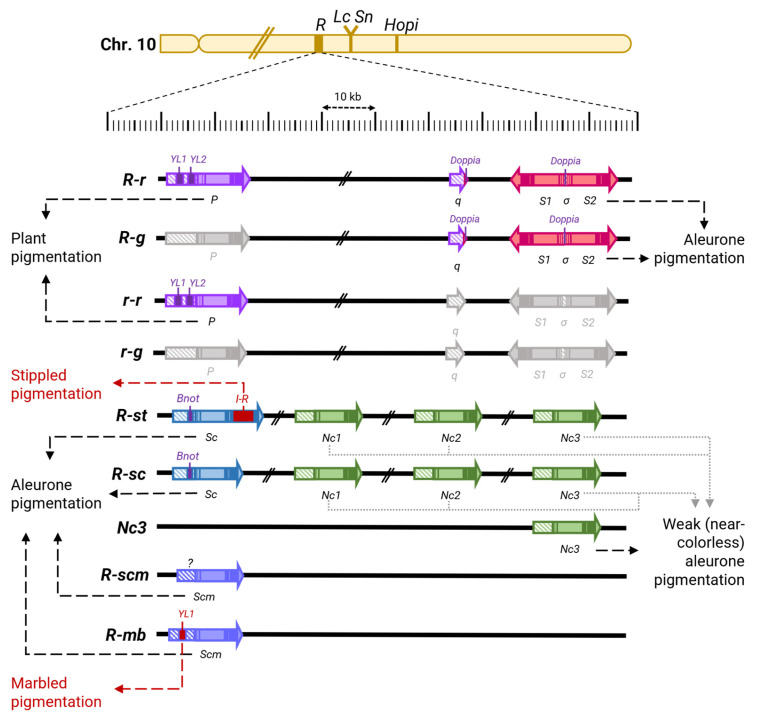
Schematic representation of the structure of some alleles of the maize (*Zea mays*) complex *R* locus. The scale bar step is 1 kb, larger marks are at 10 kb steps. Black horizontal stripes represent the long arm of maize chromosome 10 in the region of the *R* locus. The high allelic diversity of the locus is associated with multiple chromosomal rearrangements resulting in multiple copies of MYC transcription factor genes, as well as transposon insertions. Purple, pink, blue, green, and blue indicate the locations and orientations of the MYC TF genes of the *R*-family: *P*, *S1-2*, *Sc*, *Nc1-3*, *Scm*, respectively. Solid colors of gene sequences show transcribed regions, darker bands of the corresponding color show translated exons, and shaded regions show promoter regions. Gray color of *P*, *q*, *S1*, *σ*, and *S2* sequences corresponds to their non-functional variants or their absence. Component *q* is a duplicated region of the *P* gene promoter, which has a small region of homology with the *σ* region. Dark purple indicates transposons that differentiate specific genes from others. Red indicates transposon sequences present in the *Sc* and *Scm* gene sequences, which are thought to cause the stippled and marbled phenotypes, respectively. ‘?’ above the promoter of the Scm gene in R-scm indicates that the characteristic features of the gene structure in this allele are unknown. The number of *Nc* genes can vary in different derivative alleles of *R-st* and *R-sc*, but this variation has virtually no effect on their phenotype [50,52,165,201,202,203,204,205,206,207,212,216,223,224,225].

**Table 2 ijms-26-00734-t002:** Brief description of some alleles of the complex *R* locus of MYC-like bHLH genes regulating anthocyanin pigmentation in maize (*Zea mays* L.).

Allele	Structure of Allele *	Phenotype	References
*R-r* *(Red-red)*	Functional alleles of genes *S1* and *S2*; promoter of *R* gene—q; functional allele of gene *P*	Uniform anthocyanin pigmentation of aleurone (*R*, *Red*); anthocyanin pigmentation of plant organs (-*r*, -*red*)	[50,52,165,201,202,203,206,207,212,216]
*R-g*/*S* *(Red-green*/*Seed color) ***	Functional alleles of genes *S1* and *S2*; promoter of *R* gene—q; non-functional allele of gene *P*, or absence of *P*	Uniform anthocyanin pigmentation of aleurone (*R*, *Red*); anthocyaninless plant organs (-*g*, -*green*)	
*r-r*/*P* *(red-red*/*Plant color) ***	Non-functional alleles of genes *S1* and *S2*; promoter of *R* gene—q, or their absence; functional allele of gene *P*	Uncolored aleurone (*r*, *red*); anthocyanin pigmentation of plant organs (-*r*, -*red*)	
*r-g* *(red-green) ***	Non-functional alleles of genes *S1* and *S2*; promoter of *R* gene—q; non-functional allele of *P*; or their absence	Uncolored aleurone (*r*, *red*); anthocyaninless vegetative organs (-*g*, -*green*)	
*R-nj*/*Nj* *(Navajo aleurone color)*	Functional allele of gene *Nj*	Anthocyanin pigmentation of aleurone in the grain apex; anthocyanin pigmentation of plant organs	[50,165,205,223]
*R-st* *(Red-stippled) ***	Functional allele of *Sc* (*Self-coloured*) gene with insertion of mobile element *I-R* (*Inhibitor of R*) into exon 7; from 0 to 3 functional or non-functional alleles of *Nc* (*Near-colorless*) genes	Small spots of colored aleurone on a white background with variable anthocyanin pigmentation, increasing with a rise in the number of functional copies of the *Nc* genes; variable anthocyanin pigmentation of plant organs	[50,165,204,205,206,224,225]
*R-sc* *(self-coloured aleurone) ***	Functional allele of *Sc* gene (*Self-coloured*); from 0 to 3 functional or non-functional alleles of *Nc* genes	Uniform anthocyanin pigmentation of aleurone; variable anthocyanin pigmentation of plant organs	
*Nc3* *(Near-colorless 3)*	Functional allele of *Nc3* gene	Uniform weak anthocyanin pigmentation of aleurone; lack of pigmentation of plant organs	
*R-mb* *(marbled color)*	Functional allele of *Scm* (*Self-coloured marbled*) gene with insertion of *YL1* mobile element into promoter	“Marbled” anthocyanin pigmentation of aleurone with large spots of anthocyanin pigmentation on a white background of plant organs	[50,165,205]
*R-scm* *(self-coloured marbled)*	Functional allele of *Scm* (*Self-coloured marbled*) gene	Strong uniform anthocyanin pigmentation of aleurone, scutellum and plant organs	

A schematic representation of the structure of alleles is shown in Figure 4. * The structure of an allele is defined as the number and the allelic states of the MYC-encoding genes of the *R* family within the locus, as well as other genetic elements that influence their function. ** In these cases, a locus allele is understood to represent a whole group of alleles that share similar functions and phenotype, are derived from a single base allele, and have a variable structure of their constituent components.

Similar to *R*, many alleles of the *B* locus on chromosome 2 have been described, which are characterized by a wide range of phenotypic manifestations [25,54,156,167,213,214,226]. However, unlike *R*, the *B* locus contains a single copy of the MYC protein gene, and such diversity is caused exclusively by insertions of various transposable elements in the 5′-upstream region (5′ UTR) of the gene (Table 3 and Figure 5). Thus, a feature of the *B-Peru* and *B-Bolivia* alleles is that, in addition to plant organs, they activate the expression of anthocyanin biosynthesis structural genes in embryonic tissues such as the aleurone and scutellum. This activation is facilitated by a large insertion in the 5′ UTR of the gene, probably mediated by the movement of a transposon element [25,54,226]. Moreover, in *B-Bolivia*, in the region of this insertion, there is an additional, even larger insertion that has homology with retrotransposon sequences, leading to a decrease in gene activity in the embryo (Figure 5) [25,54].

**Figure 5 ijms-26-00734-f005:**
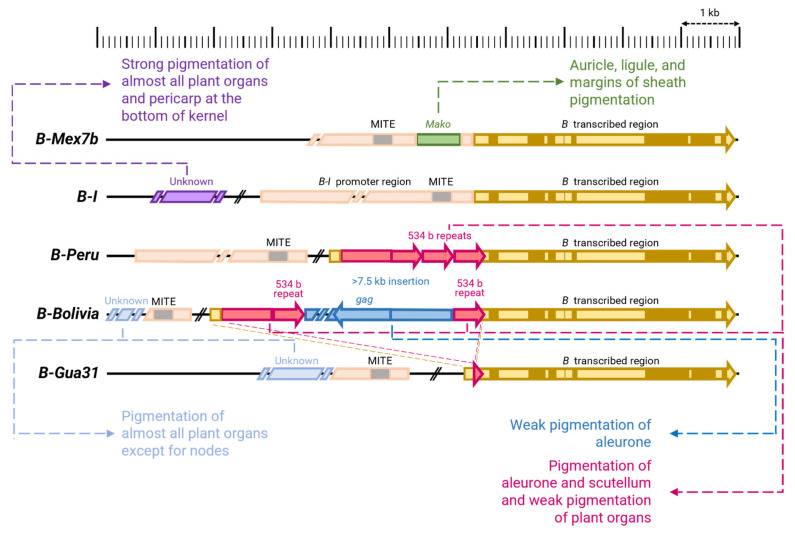
Schematic representation of the structure of some alleles of the maize (*Zea mays* L.) B locus. The scale bar step is 100 bp, larger marks are located at 1 kb steps. The transcribed region of the MYC transcription factor gene B, which has high sequence similarity in all alleles, is shown in yellow; the translated exons are shown as darker stripes. Unlike locus R, locus B contains a single copy of the MYC protein gene; the large phenotypic variation of its alleles is caused by insertions of mobile elements in the upstream region of the gene. Homologous sequences in different alleles are shown in the same colors. Green indicates the Mako element, a direct repeat-flanked sequence, probably of transposon origin, insertion in the promoter region of the B-Mex7b, which results in the reduced pigmentation, restricted to the auricles and margins of leaf sheaths [54]. Purple indicates an unknown regulatory element located far upstream of the transcription initiation site of the gene, which results in the highly pigmented phenotype of the B-I allele [54]. The dark pink color indicates the 2.5 kb insertion that causes pigmentation of embryonic tissues in the B-Peru and B-Bolivia alleles. Arrows indicate 534 bp tandem repeats within this insertion, probably of transposon origin. The dark pink arrowhead in the B-Gua31 allele sequence displays similar repeats to those limiting the 2.5 kb insertion, indicating that the allele originated through excision of this insertion, which caused the corresponding loss of grain pigmentation. The blue color indicates a long (more than 7 kb) insertion located in the B-Bolivia allele between two 534 bp repeats, one of which is lost. The arrow indicates a region homologous to the gag gene of the retrotransposon polyprotein. The blue color indicates an unknown regulatory element of the B-Bolivia allele responsible for anthocyanin pigmentation of plant organs. The identical phenotype of B-Bolivia and B-Gua31 suggests that this element is the same in B-Gua31 and that B-Gua31 originated from B-Bolivia [25,54]. The MITE (miniature inverted-repeat transposable element) element, present in all studied alleles of the B locus, is shown in gray [54,227].

**Table 3 ijms-26-00734-t003:** Brief description of some alleles of the MYC-like bHLH-encoding gene B regulating anthocyanin pigmentation in maize (*Zea mays* L.).

Allele	Structure of Allele *	Phenotype	References
B-I (Intense color)	An unknown far upstream regulatory element is present, causing strong expression	Strong anthocyanin pigmentation of leaf sheaths, auricles, ligula, stem, panicle, cob husks and pericarp at the bottom of the kernels and weak pigmentation of leaves	[54,168,214,226]
B-Mex7b	The 5′UTR of the gene is similar in sequence to B-I, but an insertion of the Mako element, presumably a transposon, is present 220 bp upstream of the start codon, leading to weakening of gene expression	Anthocyanin pigmentation of auricles, ligula and margins of leaf sheaths	[54]
B-Peru	A 2.5 kb insertion in the 5 ′UTR with three tandem repeats of 534 bp results in the development of bright pigmentation in embryonic tissues	Weak anthocyanin pigmentation of nodes, stems, spotted pigmentation of leaf sheaths, panicles, ears, strong pigmentation of aleurone and scutellum	[25,54,156,167,168,214,226]
B-Bolivia	The same as in B-Peru, insertion with an additional insertion of a large (more than 7.5 kb) element with the retrotransposon sequence gag between two 534 bp repeats with the loss of the third one, which leads to weakening of the pigmentation of the embryonic tissues. An additional unknown regulatory element is present, promoting the pigmentation of plant organs.	Anthocyanin pigmentation of stems, leaf sheaths, panicles, cob sheaths, weak pigmentation of aleurone and scutellum	[25,54,214]
B-Gua31	Presumably, the same as in B-bolivia, unknown regulatory element is present, promoting the pigmentation of plant organs	Anthocyanin pigmentation of stems, leaf sheaths, panicles, and cob husks	[25,54]

A schematic representation of the structure of alleles is shown in Figure 5. * The structure of an allele refers to the characteristic features and components of the promoter region of the gene.

The *IN1* (*Intensifier 1*) gene is interesting because it codes for MYC protein, which represses anthocyanin biosynthesis in the aleurone. Recessive mutants for this gene have a very pronounced aleurone coloration [219]. It was shown that *IN1* does not affect the expression of *B* and *C1* genes, but modulates the activity of their protein products, probably competing with the MYC protein for attachment to the complex, while preventing its normal functioning due to a defective structure, which is a consequence of incorrect splicing of most transcripts of its gene [26,219].

Among the R2R3-MYB genes of maize transcription factors regulating anthocyanin biosynthesis, it is worth noting the already mentioned genes *C1* [16,22] and *Pl* [47]. They have the opposite expression pattern in the plant [22,218]. *C1* is expressed only in the anthers [228] and embryonic tissues—the aleurone and scutellum [22], where its product, probably forming complexes with Sn, Hopi or R proteins and the WDR protein PAC1 causes anthocyanin pigmentation [15,16,23,26,28,47,52,54,126,156,157,169,220,221,228]. Notably, one of the alleles of the *C1* gene, *C1-I* (*C1 Inhibitor*), act as a dominant inhibitor of anthocyanin pigmentation, similar to *IN1*, due to the defective structure of the encoded protein C-terminal region [220,229,230], which constitutes an activator domain in the C1-like proteins [231]. The presence of such MYC and R2R3-MYB dominant repressors in maize provides an excellent opportunity for studying the structure and functional activities of MBW complexes.

In contrast to *C1*, *Pl* is expressed in many tissues and organs of maternal origin, including virtually all vegetative organs and the pericarp [218,228], where it apparently activates anthocyanin biosynthesis together with the *Sn*, *Lc*, *Hopi* genes and various *B* and *R* alleles [15,23,24,25,28,47,52,121,126,156,167,169,175,210,217,218,232]. However, B. Zhang and colleagues [28] did not find evidence for protein-protein interactions between B and C1 or Pl, which casts doubt on the possibility of the formation of B–C1 and B–Pl complexes [19,28]. This ambiguity requires further research. Interestingly, one of the alleles of the *Pl* locus, *Pl-bol3*, unlike the others, is characterized by the presence of three copies of MYB encoding genes, differing in their regulatory regions, which leads to a phenotype that is different from other alleles, in which anthocyanin pigmentation is observed in a different set of plant organs, and is also light-dependent [232]. Remarkably, in maize the *C1* and *Pl* genes, in addition to activating anthocyanin biosynthesis, represses the structural genes for the biosynthesis of flavones and, surprisingly, benzoxazinoids (specifically, DIMBOA) [228]. This indicates that, at least in cereals, the MBW regulatory network may be involved in the regulation of not only flavonoid biosynthesis, but also the biosynthesis of other secondary metabolites.

Two more duplicated MYB genes, *P1* and *P2* located in the *P* locus [64,129,131], are unique because their products, belonging to subgroup 7, cause pigmentation associated with the accumulation of phlobaphenes without the participation of a MYC partner (Figure 3A) [64,128,129,130]. These proteins bind to the promoters of early structural genes of flavonoid biosynthesis, in particular *A1* (encoding dihydroflavonol 4-reductase), via AC-rich elements in their promoters [121,143,157,175,177]. Similar to the P proteins that bind to these elements without the MYC protein, C1 is also capable of activating structural gene expression via AC-rich elements independently, although with very low efficiency. This activity of C1 is not affected by the presence of R/B proteins, but formation of the R–C1 complex results in increased *A1* expression and also allows for the activation of *Bz1* expression (encoding anthocyanidin 3-O-glucosyltransferase) via R/RIF1-mediated G-box binding (Figure 3B) [121,139,143,157,175]. Thus, MYB proteins of subgroup 7 specialize in activating the expression of early genes of flavonoid biosynthesis independently of MYC transcription factors, but are not capable of activating late genes, probably due to the necessity of the MYC-mediated G-box binding for such activation (Figure 3A,B) [121,139,177]. At the same time, they complement the functions of proteins of subgroup 5 (C1/Pl), which, in addition to such MYC-independent activity, in complexes with R/B proteins, are capable of effectively activating both early and late biosynthetic genes. For the similar subgroup 7 MYB transcription factor of rice, OsP1, specialization in MYC-independent activation of expression of early flavonoid biosynthetic genes complementing the function of the MBW complex has also been shown [132]. Interestingly, ZmP1 and OsP1 have a disrupted [D/E]Lx2[R/K]x3Lx6Lx3R motif, which is required for binding MYC transcription factors [20]. Conserved Leu and Arg in these proteins are replaced by Tre and His, respectively, which is the main reason why they are unable to bind MYC proteins [20,126]. In maize, like the genes encoding MBW complex-forming R2R3-MYB proteins, the two *P* paralogs also have spatial specificity in their function. *P1* is responsible for the pigmentation of the pericarp, panicle, cob and silks, while *P2* has a different expression pattern in the plant and activates the biosynthesis of phlobaphenes in the anthers and silks [46,128,131,222]. Similarly to *ZmR* and *OsPl*, the complex structure of the maize *P* locus allows for distinct combinations of organ and tissue coloration in the plant, depending on the alleles of the *P1* and *P2* genes [131,223]. Interestingly, one of the alleles of the locus, *P-wr* (*white pericarp and red cob*), contains six tandemly repeated copies of the *P1* gene, which leads to the absence of pigmentation of some organs, probably due to disturbances in the regulatory regions [131,233].

Currently, only one gene of the WDR transcription factor of maize that determines anthocyanin pigmentation is known—*PAC1* [26,164]. The gene is expressed in all tissues of the plant, but its mutations lead to the loss of coloration only in the root and aleurone, which suggests the presence of its paralogs with similar functions [26]. However, the second known gene of the WDR protein of maize, *ZmMP1*, is not involved in the regulation of anthocyanin biosynthesis and no interaction with MYC and MYB proteins has been shown [26,28]. Whether this fact is the evidence of the presence of other WDR genes in *Z. mays* that form complexes with R/B and C1/Pl proteins, or whether MB complexes can effectively activate structural biosynthesis genes in maize without a WDR partner is not yet known for certain. The second option can be assumed due to, for example, the ability of MYC proteins, particularly ZmR and AtGL3, to form homodimers through ACT domains with the participation of EMSY-like proteins such as ZmRIF1 [139,157,174,176], which can represent a WDR-independent regulatory system, or due to other mechanisms of MB complex formation briefly described above (Figure 2f,h,i). However, it has been shown that complexes formed by dimerization through ACT domains are capable of activating the expression of only some structural genes [176], which, together with the fact of ubiquitous expression of the *PAC1* gene [26], contradicts this hypothesis. This poses interesting questions for researchers, the answers to which will lead to a better understanding of the mechanisms of anthocyanin biosynthesis regulation by MB/MBW complexes.

#### 4.4.2. Diversity of MBW Genes in Rice (*Oryza sativa* L.)

In rice, a variety of MYC-encoding genes determining flavonoid accumulation and anthocyanin pigmentation has been described (Table 4). Among these, the *OsPl* (*Purple leaf*) locus has a wide variation in phenotypic manifestations similar to *ZmR* and *ZmB* [53,234,235], associated with the complex structure of its sequence. It includes two genes that have high homology to the *R*/*B* family genes, *OsB1* (other names are *Ra1*, *Red a1* [236], *Pb*, *Purple bran* [237] and *Ps*, *Purple stigmas* [238]) and *OsB2* (other names are *Kala4*, *Key gene for black coloration by anthocyanin accumulation on chromosome 4* [181] and *S1* [44]). Different combinations of alleles of these two genes result in different allelic states of the *Pl* locus [53]. The phenotypes of the functional alleles of the *Pl* locus are characterized by different combinations of anthocyanin pigmentation of leaf blades, sheaths and auricles, ligula, nodes, internodes, pericarp, stigma, and glumes [53,235,238]. *OsB1* has 11 exons and 10 introns, the arrangement of which strictly corresponds to those of *ZmLc* [53]. The gene is mapped to a chromosomal region that has synteny with chromosomes 2 and 10 of maize, where the *B* and *R* genes are located. These data indicate a common origin of these genes in different species of cereal plants [53,236]. *OsB1* (*Ra1*, *Pb*, *Ps*) is a key gene that determines the anthocyanin pigmentation of the stigma, and its product forms a complex with R2R3-MYB partner OsC1 [238]. An interesting feature of the *OsB2* gene is the ability of its product to activate the expression of its MYB partner gene *OsC1* alone, and in combination with it to activate the expression of the WDR gene *OsPAC1* (*OsTTG1*, *WA1*), which is described above [44]. It has been shown that the third gene of the MYC transcription factor, *OsPa* (*Purple apiculi*), is located in the *OsB1* and *OsB2* gene cluster. OsPa together with OsC1 activates the biosynthesis of anthocyanins in the tips of spikelet glumes [238,239]. Other genes encoding MYC regulators of flavonoid biosynthesis in rice have also been identified. The *OsPSH1* (*Purple leaf sheath 1*) locus, containing two paralogs *OsRb1* and *OsRb2*, together with the *OsC1* cause the purple pigmentation of leaf sheaths [49,240]. The *OsRc* gene causes the accumulation of proanthocyanidins in the outer grain coats [235,241]. It is interesting that the *OsRc* gene is an ortholog of the maize negative regulator of anthocyanin biosynthesis *IN1* and among the MYC genes of *Z. mays* it has the greatest sequence similarity with it. However, *OsRc* has the opposite function—it activates the flavonoid biosynthesis pathway [241].

Among the identified R2R3-MYB transcription factor genes in rice are *Kala3* [242], *OsPL* (*Purple Leaf*) [243], *OsPL6* (*Purple Leaf 6*) [244] and *OsP1* [132]. For *OsP1*, it was shown that, similar to the highly homologous *ZmP1*, its product, belonging to the subgroup 7 of R2R3-MYB proteins, activates the expression of early genes of flavonoid biosynthesis independently of MYC proteins, thereby regulating anthocyanin pigmentation of leaves (Figure 3A) [132]. The Kala3 protein forms a complex with OsB2 (Kala4/S1) to activate the anthocyanin biosynthesis pathway in the pericarp. An interesting property of this gene is that it encodes a protein that combines the characteristics of R2R3-MYB transcription factors of both subgroups 5 and 6, which has also been observed in regulators of anthocyanin pigmentation from some orchids and onions [127]. Some evidence suggests that the Kala3 protein may be involved in the regulation of proanthocyanidin biosynthesis, probably in association with the MYC protein OsRc [245]. A significant level of *OsPL* expression is observed in leaf blades, sheaths and stems. Plants carrying a functional allele of this gene have purple anthocyanin pigmentation of these organs [243]. *OsPL6* is a positive regulator of anthocyanin biosynthesis in leaves, including the midrib [244].

The above-mentioned *OsC1* gene coding R2R3-MYB transcription factor, named by analogy with the highly homologous maize *ZmC1* gene, is also well studied [246]. Its product is capable of forming MBW complexes with many MYC proteins of the R/B family and the WDR protein OsPAC1 (OsTTG1, WA1, WD40 for Anthocyanin biosynthesis 1), which together regulate the pigmentation of almost all plant tissues and organs, with the exception of at least the pericarp [44,53,132,231,238,239,240,247,248]. It was assumed that *OsC1* is able to directly, without a MYC partner, activate flavonoid biosynthesis in glumes and pistil stigmas, causing brown pigmentation (as opposed to red, which appears in the presence of a functional allele of the MYC protein gene [246,248]). However, recent studies have shown that this was erroneous; brown pigmentation was probably caused by the absence of a functional dihydroflavonol 4-reductase gene [238], which leads to the accumulation of brown flavonols and flavanones [248]. Thus, *OsC1* causes the accumulation of flavonoid compounds in various tissues and organs in complexes with *R*/*B* proteins, which determine the spatial specificity of this process.

The WDR ortholog of the *ZmPAC1* and *AtTTG1* genes, which regulates anthocyanin biosynthesis in many plant tissues—*OsPAC1* (other names are *OsTTG1* and *OsWA1*) [44,132,249]), is well characterized in rice. Mutations in the gene result in the absence of anthocyanin pigmentation in roots, leaf blades and sheaths, spikelets, glumes, and pistil stigmas, and in weakening of the color of the stem, ligula, and grain coats [249]. The OsPAC1 protein has been shown to form MBW complexes with at least the OsC1 and OsB2/OsRb/OsRc proteins [44,132,240,249]. Interestingly, its expression in glumes is activated by the OsB2–OsC1 complex (Figure 3F(c,d)) [44].

Some evidence suggests that, although *OsB1*, *OsB2*, *OsC1*, *Kala3*, and *OsPAC1* are the key MBW regulatory genes of anthocyanin biosynthesis in rice, other MBW genes, such as *OsbHLH2* (*LOC_Os04g47040*), *MYBs* (*LOC_Os01g49160*, *LOC_Os01g74410*, and *LOC_Os03g29614*) [250], *OsiWD40-3* and *OsiWD40-15* [249,251], may also be involved in this process.

Thus, the formation of classical MBW complexes regulating flavonoid biosynthesis has been described in rice, but the regulatory networks formed by MYC, MYB, and WDR regulators appear to differ from those described in dicotyledons (see Figure 3). A brief description of the above-described genes of rice MBW proteins is presented in Table 4.

**Table 4 ijms-26-00734-t004:** Brief description of characterized loci and genes of MYC-like bHLH, R2R3-MYB, and WDR regulators of anthocyanin pigmentation in rice (*Oryza sativa* L.).

Family	Gene (Locus)	Chromosome	Function	MBW Complex Formation	Note	References
MYC-like bHLH	*OsPl* (*Purple leaf*)	4	Positive regulator of anthocyanin pigmentation of leaf blades, sheaths and auricles, ligula, nodes, internodes, pericarp, stigma and glume tips, depending on the allele	Yes, with OsC1 and OsPAC1	Contains two *MYC* genes with different expression patterns—*OsB1* and *OsB2* (*Booster 1, 2*). *OsB2* activates the expression of its MYB partner gene *OsC1*, and in combination with it, activates the expression of the WDR partner *OsPAC1*	[53,181,234,235,236,238,248]
	*OsPa* (*Purple apiculi*)	4	Positive regulator of anthocyanin pigmentation of glume tips	Yes, with OsC1 and, probably, OsPAC1	**-**	[238,239]
	*OsPSH1* (*Purple leaf sheath 1*)	1	Positive regulator of anthocyanin pigmentation of leaf sheaths	Yes, with OsC1 and OsPAC1	Contains two *MYC* genes, *OsRb1* and *OsRb2*, that jointly regulate anthocyanin biosynthesis in leaf sheaths	[236,240]
	*OsRc* (*Red c*)	7	Positive regulator of proanthocyanidin biosynthesis in outer grain hulls	Possibly, with Kala3	Ortholog of *ZmIN1*, however, performs the opposite function.	[235,241,252]
R2R3-MYB	*OsC1* (*Colorless 1*)	6	Positive regulator of anthocyanin pigmentation of almost all tissues and organs, except pericarp	Yes, with R/B family and OsPAC1	Expression is activated by OsB2, which, in combination with it, activates the expression of the WDR partner *OsPAC1*	[53,132,238,239,240,246,247,248]
	*OsKala3 * (*Key gene for black coloration by anthocyanin accumulation on chromosome 3*)	3	Positive regulator of anthocyanin pigmentation of pericarp	Yes, with OsB2 and, probably, OsPAC1	Combines the properties of R2R3-MYB transcription factors of both subgroups 5 and 6	[127,181]
	*OsPL* (*Purple Leaf*)	5	Positive regulator of anthocyanin pigmentation of leaf blades, sheaths and stems	Unknown	**-**	[243]
	*OsPL6* (*Purple Leaf 6*)	6	Positive regulator of anthocyanin pigmentation of leaf blades, including leaf midrib	Unknown	**-**	[244]
	*OsP1* (*Pericarp color 1*)	3	Positive regulator of anthocyanin biosynthesis in leaves	No	Activates the expression of early genes of flavonoid biosynthesis independently of MYC proteins	[132]
WDR	*OsPAC1/TTG1/WA1* (*Pale aleurone color 1/Transparent Testa Glabra 1/WD40 for Anthocyanin biosynthesis*)	2	Positive regulator of anthocyanin pigmentation of almost all plant tissues	Yes, with R/B family and OsC1	Expression is activated by the OsB2-OsC1 complex	[44,132,240,249]

#### 4.4.3. Diversity of MBW Genes in Barley (*Hordeum vulgare* L.)

Several MBW regulators of flavonoid biosynthesis are known in barley (Table 5). Its known R2R3-MYB flavonoid biosynthesis regulators are encoded by the following genes: *HvANT1* (*Anthocyaninless 1*, other names are *HvRs*, *Red stem*, and *HvMpc1*, *Myb protein c1* [253,254]), *HvMpc1-2* (*HvMpc2*) [255]), *HvMpc1-3* [56], and *HvANT28* (*HvMpc10*) [256]. Recessive mutants for *HvANT1* are characterized by the absence of pigmentation of the stem, auricles, awns and lemma [253,257]. The gene also regulates the purple coloration of leaf sheaths [254] and pericarp [258]. It is orthologous to the wheat homoeologs *TaPp-1*/*TaRc-1*/*TaC1*/*TaMpc1-1* (*Purple pericarp 1/Red coleoptile 1/Colorless 1*), which regulate anthocyanin pigmentation of coleoptile and pericarp [253,258,259]. As well as in wheat it is located on the short arm of chromosome 7.

*HvMpc1-2* (*HvMpc2*) controls anthocyanin pigmentation of the lemma and aleurone, its expression is also detected in the stems and, to a lesser extent, in the pericarp [56,255]. *HvMpc1-3* likely complements the function of *HvMpc1-2* in the aleurone, where high levels of its expression are found in anthocyanin-pigmented plants [56]. Transcripts of the gene are also detected in small amounts in the lemma, pericarp and stems, which does not correlate with their color. Interestingly, the expression of *HvMpc1-3* is noticeably lower in the anthocyanin-pigmented lemma than in the colored one, which suggests an interaction of MYC transcription factors, in which the products of the lemma pigmentation regulator genes *HvMpc1-2* and, probably, *HvANT1*, suppress the expression of *HvMpc1-3* [56]. Perhaps the difference in interactions, in which *HvMpc1-2* and *HvMpc1-3* are co-expressed at a high level in the blue-pigmented aleurone, while *HvMpc1-2* suppresses the expression of *HvMpc1-3* in the violet-colored lemma [56], are caused by the formation of different MBW complexes. It has been shown that HvMpc1-2 regulates aleurone pigmentation in conjunction with the HvMyc2 protein [75,255], while the only known MYC regulator of lemma pigmentation is HvANT2 [260]. Although the interaction of HvMpc1-2 and HvANT2 has not been studied, it can be assumed that they are able to interact, with the HvMpc1-2–HvMyc2 protein complex positively regulating the expression of *HvMpc1-3* in the aleurone, while the alternative HvMpc1-2–HvANT2 complex formed in the lemma, on the contrary, suppresses the expression of this gene. Another possibility is that HvMpc1-2, while not associated with MYC, directly inhibits the expression of *HvMpc1-3* in the lemma and triggers the biosynthesis of purple anthocyanins. In contrast, when in a complex with HvMyc2, it activates the accumulation of *HvMpc1-3* product and together with it leads to the biosynthesis of blue anthocyanins. This issue is of great interest from the point of view of mutual regulation of MYB transcription factors and requires further research.

*HvANT28* (*HvMyb10*) controls proanthocyanidin biosynthesis in grain hulls and is an ortholog of the *TaR-1* (*TaMyb10*) genes. It encodes a protein evolutionarily close to *Arabidopsis* regulator of proanthocyanidin biosynthesis *AtTT2* [62,256]. It is assumed that *Myb10* genes can trigger the expression of structural genes of flavonoid biosynthesis without the participation of MYC partners, however, this has not been reliably confirmed.

MYC-encoding genes are also present in multiple copies in the barley genome. The *HvANT2* (*HvMyc1*) encodes a MYC-like bHLH TF, which belongs to the *R*/*B* family and has high homology with the *ZmR*/*B*, *OsR*/*B*, *PhAN1* and *AtTT8* genes [260]. The gene controls anthocyanin pigmentation of leaf auricles, lemma, awns and pericarp [258,260,261]. *HvANT1* and *HvANT2* exhibit recessive-epistatic interactions and jointly control plant pericarp coloration [45,258,262]. Their protein products physically interact with each other in domesticated barley [45], as well as with the HvWD40-140 protein in Qingke (Tibetan hulless barley, *H. vulgare* var. *nudum* (L.) Hook.f.) [245]. Interestingly, the expression of *HvANT1* and *HvANT2* is significantly enhanced in the presence of dominant alleles of both genes [258], and direct activation of the *HvANT2* promoter has been shown upon *HvANT1* overexpression [45]. Thus, *HvANT1* and *HvANT2* exhibit interactions similar to *AtTT2* and *AtTT8*, in which accumulation of the MYB protein TT2 promotes expression of the MYC gene *TT8* [12,45,258]. This once again confirms the presence of complex regulatory networks between MYC and MYB genes in cereal plants, similar to those described in eudicots, which help to achieve self-sustaining expression of MBW regulatory genes for rapid, time- and tissue-specific accumulation of flavonoid compounds. HvANT1, without a MYC partner, is able to activate the expression of some structural genes of flavonoid biosynthesis, in particular *ANS*, however, the manifestation of purple anthocyanin pigmentation in transgenic plants is possible only in the presence of functional alleles of both *HvANT1* and *HvANT2* genes [45].

The *HvMyc2* gene, together with *HvMpc1-2* (*HvPL1*/*HvMpc2*), regulates anthocyanin pigmentation of aleurone [255]. Interestingly, the *HvMyc2*, *HvMpc1-2*, and *HvF3′5′H* genes are closely colocalized forming the trigenic cluster designated *MbHF35* (*MYB-bHLH-F3′5′H*). This cluster is assumed to be corresponding to the *Blx1* (*Non-blue aleurone xenia 1*) locus. The *MbHF35* cluster has a common function in aleurone and determine its blue pigmentation [75,255]. Although Strygina et al. (2017) [255] suggest that *HvMpc1-2* corresponds to a different locus and that *HvF3′5′H* corresponds to the *Blx4* locus, Jia et al. (2020) [75] mapped the genes to the interval corresponding specifically to *Blx1*. They found evidence for direct interaction of HvMyc2 and HvMpc1-2 with the *HvF3′5′H* promoter and thus describe the trigenic cluster of the *Blx1* locus as a coordinately regulated unit that determines the blue aleurone trait. A similar cluster was found in the corresponding region of chromosome 4D of wheat, mapped to the *TaBa1* (*Blue aleurone 1*) locus, and *Aegilops tauschii* Coss., as well as of chromosome 7R of rye *Secale cereale* L., so the authors suggest a common origin of the trait in the Triticeae tribe, which is associated with the formation of this trigenic locus [75]. Thus, HvMyc2 and HvMpc1-2 coordinately activate anthocyanin biosynthesis in aleurone and most likely form a classical MBW complex [255].

The only known barley gene of the WDR protein regulating flavonoid biosynthesis, *Ant13/HvWD40-1*, is orthologous to *ZmPAC1*. Its expression is detected in many parts of the plant, including roots, stem, leaf sheaths, pericarp, lemma, and aleurone [255,263]. The recessive *Ant13* mutants are characterized by the absence of anthocyanins in the vegetative organs. Notably, these mutants also lack proanthocyanidins in the seeds [263]. Since it is assumed that the key regulator of proanthocyanidin biosynthesis in barley, the R2R3-MYB protein Myb10, acts independently of MYC proteins [62,256], an interesting question arises: whether this process is regulated by a potential Myb10-Ant13 MW complex, or if an unknown regulatory network exists, involving an interaction between MBW complexes and independently acting MYB proteins. So far, no evidence of direct protein-protein interactions of HvAnt13 with MYC and R2R3-MYB proteins has been found. However, the HvWD40-140 protein of Qingke was shown to physically interact with the HvANT1 and HvANT2 proteins [245]. It is highly likely that the Qingke *HvWD40-140* gene corresponds to *HvAnt13*. Thus, in barley, classical MBW complexes regulating anthocyanin and proanthocyanidin biosynthesis are formed, including HvANT1–HvANT2–HvWD40-140 (Ant13), as well as a putative HvMyc2–HvMpc1-2–Ant13 complex.

A brief description of the barley MBW transcription factor genes described above is presented in Table 5.

**Table 5 ijms-26-00734-t005:** Brief description of characterized genes of MYC-like bHLH, R2R3-MYB, and WDR regulators of anthocyanin pigmentation in barley (*Hordeum vulgare* L.).

Family	Gene (Locus)	Chromosome	Function	MBW Complex Formation	Note	References
MYC-like bHLH	*HvANT2/HvRc/HvMyc1* (*Anthocyaninless 2/Red coleoptile*)	2H	Positive regulator of anthocyanin pigmentation of coleoptile, leaf sheaths and auricles, lemma, awns and pericarp	Yes, with HvANT1 and HvAnt13	Mutually activates the expression of *HvANT1*	[45,255,258,260,261,262]
	*HvMyc2*	4H	Positive regulator of anthocyanin pigmentation of aleurone	Yes, with HvMpc1-2 and, probably, HvAnt13	Closely colocalized with *HvMpc2* and *HvF3′5′H* at the *Blx1* locus and form a coordinately regulated unit with them	[75,255]
R2R3-MYB	*HvANT28/HvR-1/HvMyb10* (*Anthocyaninless 28/Red 1*)	3H	Positive regulator of proanthocyanidin biosynthesis in outer grain coats	Possibly, no	-	[62,256]
	*HvANT1/HvMpc1-1* (*Anthocyaninless 1/MYB protein c1 1*)	7H	Positive regulator of anthocyanin pigmentation of coleoptile, leaf sheaths and auricles, lemma, awns and pericarp	Yes, with HvANT2 and HvAnt13	Mutually activates the expression of *HvANT2*	[253,254,255,257,258]
	*HvMpc1-2/HvMpc2/HvPL2* (*MYB protein c1 2*/*Purple Leaf 2*)	4H	Positive regulator of anthocyanin pigmentation of lemma and aleurone	Yes, with HvMyc2 and, probably, HvAnt13	Closely colocalizes with *HvMyc2* and *HvF3′5′H* at the *Blx1* locus and form a coordinately regulated unit with them. Likely represses *HvMpc1-3* expression in the lemma but activates it in the aleurone	[57,75,255,264]
	*HvMpc1-3* (*MYB protein c1 3*)	4H	Positive regulator of anthocyanin pigmentation of aleurone	Unknown	Expression is repressed in the lemma but activated in the aleurone. Likely is a coregulator of *HvMpc2* in the aleurone and regulated by it	[57]
WDR	*HvAnt13/HvWD40-140* (*Anthocyaninless 13*)	6H	Positive regulator of flavonoid biosynthesis in aleurone, pericarp, lemma, stem and leaf sheaths	Yes, with HvANT1 and HvANT2, and, probably, with other Mpc1 and Myc proteins	Exhibits pleiotropic effects in plant growth	[245,255,263]

#### 4.4.4. Diversity of MBW Genes in Wheat (*Triticum aestivum* L.)

Wheat exhibits a wide variety of R2R3-MYB and MYC transcription factor genes regulating flavonoid biosynthesis (Table 6).

The first identified wheat MYB encoding sequences with high homology to the *ZmC1* gene were designated *TaMpc1* [265]. The following *TaMpc1* genes have been characterized.

A series of tightly linked MYB-coding genes on the chromosome 7D controlling the pigmentation of the pericarp (*Pp-D1*), leaf blades (*Plb-D1*, *Purple leaf blade*), leaf sheaths (*Pls-D1*, *Purple leaf sheaths*), culms (*Pc-D1*, *Purple culm*), auricles (*Ra-D1*, *Red auricles*), coleoptiles (*Rc-D1*), and anthers (*Pan-1*, *Purple anthers*) have previously been described in wheat [266,267,268,269,270]. However, within the region of this gene ‘cluster’, only one gene encoding an R2R3-MYB transcription factor, *TaMpc1-D1* (*TaC1-D1*/*TaPpm1*, *Purple pericarp MYB 1*), was found [56,253,259]. Therefore, it can be assumed that this gene is characterized by a series of alleles controlling various combinations of pigmentation of the pericarp, anthers, and vegetative organs such as the coleoptile, culm, auricles, and leaf sheaths. *TaPp-1* homoeologs (*TaRc-1*/*TaMpc1-1*/*TaC1*) are orthologous to the rice *OsC1* gene and control coleoptile and pericarp coloration [56,253,259,269,271,272,273].

*TaMpc1-A2* is expressed in the coleoptile, which correlates with its anthocyanin pigmentation, and likely complements the function of *TaPp-1* (*TaRc-1*/*TaMpc1-1*/*TaC1*). *TaMpc1-D4* (*TaPL1-4D2*, *Purple Plant 1*) has a high expression level in the coleoptile and pericarp that does not, however, correlate with the pigmentation of these tissues [56]. *TaMpc1-D2* (*TaPL1-4D1*) is weakly expressed in the coleoptile, the exact function of this gene remains unknown. *TaMpc1-D3* (*TaPL1-4D3*) expression is not detected in either the coleoptile or the pericarp [56]. This gene corresponds to *TaMYB4D*, one of the genes in the *MbHF35* (*MYB-bHLH-F3′5′H*) cluster, similar to that found in barley, which also includes *TaMYC4D* (*TaMyc2.6*) and *TaF3′5′H*. The cluster is located at the *Ba1* (*Blue aleurone 1*) locus and regulates the blue anthocyanin pigmentation of the aleurone [55,75]. Strygina & Khlestkina [56] did not detect expression of any *TaPL1* (*TaMpc1-B2*, *-D2*, *-D2*, and *-D4*) genes in the coleoptile and roots, but in a previous study [264], *TaPL1* expression was detected in these organs. The authors of the later study suggest that Shin et al. [264] cloned the *TaMpc1-A2* transcript while attempting to detect *TaPL1*. Further studies are needed to establish the exact tissue specificity of *Mpc1* genes in wheat.

Strygina & Khlestkina [56] proposed new designations for the previously described *Mpc1* genes of the Triticeae tribe based on their phylogeny. According to their analysis, the *Mpc1* genes are divided into three clades: *Mpc1-1*, which are located on homeologous group 7 chromosomes and include barley *HvMpc1-1* (*HvANT1*/*HvC1*) and wheat homoeologs *TaMpc1-1* (*TaRc-1*/*TaPp-1*/*TaC1*) in the A, B, and D subgenomes [253,259,274,275]; *Mpc1-2* and *-4*, located on homeologous group 4 chromosomes, which include *HvMpc1-2* (*HvMpc2*/*HvPL1*), *TaMpc1-B2* (*TaPL1-4B1*), *TaMpc1-D2* (*TaPL1-4D1*), and *TaMpc1-D4* (*TaPL1-4D2*) [264]; and *Mpc1-3*, which are also located on group 4 chromosomes and include *HvMpc1-3* and *TaMpc1-D3* (*TaPL1-4D3*). These phylogenetic data indicate that the duplications that gave rise to these three groups of R2R3-MYB-encoding genes occurred before the divergence of the Triticeae tribe, since sequences of the three clades are found in *H. vulgare*, and various species of the genera *Aegilops* and *Triticum* [56].

Three homeologous copies of *TaR-1* (*Red-1*) were shown to also encode MYB proteins designated *TaMyb10* [62,256,276]. *TaR-1* (*TaMyb10*) homeologs are evolutionarily related to a clade of MYB proteins regulating proanthocyanidin biosynthesis, including *AtTT2* and *HvANT28* (*HvMyb10*), and are responsible for the red coloration of the outer grain coats [62]. As mentioned above, it is assumed that *Myb10* genes act independently of MYC proteins.

It has been shown that the *TaPp-3* (*Purple pericarp 3*) locus is orthologous to *OsPl* and *ZmR* and corresponds to the gene of the MYC transcription factor *TaMyc-A1* (*TaMyc1*, *TaPpb1*, *Purple pericarp bHLH 1*), the expression of which strictly correlates with the expression of structural genes for anthocyanin biosynthesis and the manifestation of purple pigmentation of the pericarp, coleoptile, leaf auricles and glumes [57,259,274,275]. The gene is capable of activating anthocyanin biosynthesis in the pericarp only in association with one of the *TaPp-1* homeologs (*TaRc-1*/*TaMpc1-1*/*TaC1*)—*TaPp-A1* or *TaPp-D1* (*TaPpm1*) [259,274,275]. At the same time, the formation of a complex by TaPp-3 (TaPpb1) and TaPp-D1 (TaPpm1) proteins has been proven. This complex activates the expression of *TaPp-D1* (*TaPpm1*) [259], which is reminiscent of the interactions of AtTT2–AtTT8 and HvANT1–HvANT2 proteins, which, by forming MBW complexes, trigger the expression of *AtTT8* and *HvANT2*, respectively [12,45,258]. However, neither TaPp-3 nor TaPp-D1 alone are capable of such regulation [259], which distinguishes them from AtTT2 and HvANT1, which are capable of activating the expression of *AtTT8* and *HvANT2*, respectively, without a MYC partner (see Figure 3) [12,45,258].

Interestingly, in plants carrying dominant *TaPp-A1* alleles, the presence of the functional *TaPp-D1* (*TaPpm1*) allele significantly reduces the expression level of *TaPp-3* (*TaPpb1*), but at the same time enhances anthocyanin pigmentation [175]. Moreover, in such plants the expression of another *Mpc1* gene *TaMpc1-D4* (*TaPL-4D2*) is significantly increased [56]. A possible explanation for this fact is the formation of the TaPp-3–TaPp-D1 complex, which is incapable or has low efficiency in activating the expression of *TaPp-3*, but is capable of effectively activating the biosynthesis of anthocyanins, while the potential TaPp-3–TaPp-A1 complex, on the contrary, activates *TaPp-3*, but do not activate structural genes. The interaction of *TaPp-3* and *TaPp-D1* products is very likely, since bright pericarp coloration is observed only in plants carrying dominant alleles of these genes [270,275]. In this case, the formation of TaPp-3–TaPp-D1 should lead to a smaller number of TaPp-3–TaPp-A1 complexes and a corresponding decrease in *TaPp-3* expression. Further studies of these three genes are of considerable interest and should improve the understanding of the interaction of different MBW complexes.

Other genes, highly homologous to *TaPp-3* (*TaPpb1*), have also been characterized: *TaMyc-A2* (*TaMyc1.2*), *TaMyc-B1* (*TaMyc1.3*), *TaMyc-D1* (*TaMyc1.5*), *TaMyc-D2* (*TaMyc1.4*), which, together with *HvANT2* (*HvMyc1*), are grouped into a separate clade of MYC transcription factors of the Triticeae tribe [55,57,275]. All genes, except *TaMyc-D1* (*TaMyc1.5*), which is non-functional, are expressed to varying degrees in the coleoptile, but only for *TaMyc-A2* and *TaMyc-B1* a correlation between expression and anthocyanin pigmentation of this tissue was shown [57]. *TaMyc-B1* is likely a regulator of coleoptile and pericarp anthocyanin pigmentation, where it complements the function of TaPp-3–TaPp-1 [57].

As mentioned above, the *TaMpc1-A2* and *TaMpc1-D4* genes are also expressed in the coleoptile and pericarp [56]. The authors suggest that TaMpc1-A2 and TaMpc1-D4 are coregulators of the TaPp-1–TaPp-3 complex and may determine the biosynthesis of uncolored anthocyanin precursors [56], substituting TaPp-1 [56]. Another possibility is that another MYC partners, different from TaPp-3 and which functional alleles were absent in the lines studied by Strygina and Khlestkina [56] may be active in the pericarp and coleoptile and thus provide pigmentation of these organs by forming complexes with TaMpc1-A2 и TaMpc1-D4. Such a partner of the *TaMpc1-D4* gene may be *TaMyc-A2*, the expression of which in the weak pigmented coleoptile of the line S29 (Saratovskaya 29) is significantly higher than in other lines with uncolored and brightly colored coleoptile [57]. S29 carries recessive alleles of *TaPp-D1* and *TaPp-3*, while it also has a high level of expression of *TaMpc1-D4* [56,57]. Perhaps, the interaction of *TaMyc-A2* and *TaMpc1-D4* is the cause of weak coleoptile pigmentation in S29 plants, while intense anthocyanin pigmentation in other lines is due to the activity of *TaPp-1*, *TaPp-3* [57,259,275], and, probably, *TaMyc-B1* [57]. Interestingly, the expression of *TaMyc-B1* is significantly increased in the presence of the dominant *TaPp-A1* allele in the plant, which suggests a joint regulation of coleoptile coloration by these two genes, similar to *HvANT1–HvANT2* and *AtTT2–AtTT8* systems, where *MYC* gene expression is triggered through the MYB transcription factor [57], which can be expected given the close relationship of these genes in wheat and barley [55,56,57].

The second clade of the Triticeae *Myc* genes, together with *HvMyc2*, consists of the chromosome 4 genes *TaMyc2.1* in the A genome, *TaMyc2.2* and *TaMyc2.3* in the B genome, and *TaMyc2.4*, *TaMyc2.5*, and already mentioned *TaMyc2.6* (*TaMYC4D*) in the D genome [55]. Although the functional role of the *TaMyc2.1*-*TaMyc2.5* genes has not been studied, the close relationship with *HvMyc2* suggests that at least some of them control anthocyanin pigmentation. A brief description of the above-described MYC and MYB genes of wheat transcription factors with known functions is presented in Table 6.

In wheat, the gene (or genes) of the WDR transcription factor regulating flavonoid biosynthesis have not yet been identified. However, the formation of a complex by the TaPp-3 and TaPp-D1 proteins, possible interactions of TaPp-3–TaPp-A1 and TaMyc-A2–TaMpc1-D4, as well as the high homology of these factors with the above-discussed MYC and MYB regulators of maize, rice, and barley allow us to confidently assume the presence of classical MBW complexes in wheat. The great diversity of MYC and MYB genes in hexaploid wheat resulting from the combination of three genomes, makes the study of its MBW complexes particularly interesting and important in terms of the specialization of evolutionarily close MYC and MYB regulators. The spatio-temporal specificity and complex system of mutual regulation of transcription factors such as TaPp-3, TaMyc-D1, TaPp-A1, TaMyc-A2, TaPp-B1, and others [56,57,259,275] presents an excellent opportunity to study the interactions of different MBW proteins and their specialization in certain functions.

In summary, plants possess multiple copies of the MYC-like bHLH and R2R3-MYB transcription factor genes, the products of which control the anthocyanin pigmentation of various tissues and organs. The presence of numerous functional copies of these regulatory genes allows plants to precisely regulate the accumulation of anthocyanins in their different parts. At the same time, it is uncertain whether the WDR proteins, involved in the regulation of anthocyanin biosynthesis, are the same for all MBW complexes, or whether they are also characterized by a diversity that contributes to the formation of different types of complexes with different functions. Further study of the known MYC, MYB and WDR encoding genes and the discovery of new ones should expand our knowledge of the functioning and relationships of these transcription factors.

**Table 6 ijms-26-00734-t006:** Brief description of characterized genes of MYC-like bHLH and R2R3-MYB regulators of anthocyanin pigmentation in wheat (*Triticum aestivum* L.).

Family	Gene (Locus)	Chromosome	Function	MBW Complex Formation	Note	References
MYC-like bHLH	*TaPp-3/TaMyc-A1/TaMyc1.1/TaPpb1* (*Purple pericarp 3/Purple pericarp bHLH 1*)	2A	Positive regulator of anthocyanin pigmentation in pericarp, coleoptile, leaf auricles and glumes	Yes, with TaPp-D1 and, probably, TaPp-A1 and TaPp-B1	Expression is higher in the presence of the functional allele of *TaPp-A1* than in the presence of both *TaPp-A1* and *TaPp-D1*. In complex with TaPp-D1 activates expression of *TaPp-D1*.	[258,259,270,274,275]
	*TaMyc-A2/TaMyc1.2*	2A	Probably a positive regulator of anthocyanin pigmentation in the coleoptile	Probably, with TaPp-1 and TaMpc1-D4	Probably, is coregulator of TaPp-1-TaPp-3 complexes, potentially activating the biosynthesis of uncolored anthocyanin precursors in association with TaMpc1-D4	[55,57]
	*TaMyc-B1/TaMyc1.3*	2B	Probably a positive regulator of anthocyanin pigmentation in the coleoptile	Probably, with TaPp-1	Expression is increased in the presence of the functional allele of *TaPp-A1*. Potential coregulator of TaPp-3	[55,57]
	*TaMYC4D/TaMyc2.6*	4D	Potential positive regulator of anthocyanin pigmentation of aleurone	Possibly, with TaMYB4D	Closely colocalizes with *TaMYB4D* and *TaF3′5′H* at the *Ba1* (*Blue aleurone 1*) locus	[75,255]
R2R3-MYB	*TaR-1/TaMyb10* (*Red 1/MYB protein c 10*)	3	Positive regulator of proanthocyanidin biosynthesis in outer grain coats	Possibly, no	-	[62,256,276]
	*TaPp-A1/TaRc-A1/TaMpc1-A1/TaC1-A1* (*A genome Red coleptile 1/Purple pericarp 1/MYB protein c1-1/Colorless 1*)	7A	Positive regulator of anthocyanin pigmentation in pericarp and coleoptile	Probably, with TaPp-3 and TaMyc-B1	Increases *TaMyc-B1* expression in coleoptile. Probably shares redundant functions with TaPp-D1.	[56,253,258,259,269,270,272,274,275]
	*TaPp-B1/TaRc-B1/TaMpc1-B1/TaC-B1* (*B genome Purple pericarp 1/Red coleptile 1/MYB protein c1-1/Colorless1-1*)	7B	Potential positive regulator of anthocyanin pigmentation in coleoptile and pericarp	Probably, with TaPp-3	Expression significantly increases in the presence of the dominant *TaPp-A1* allele. Potential coregulator of TaPp-A1 and TaPp-D1.	[56,253]
	*TaPp-D1/TaRc-D1/TaMpc1-D1/TaC-D1/TaPpm1* (*Purple pericarp MYB 1*)	7D	Positive regulator of anthocyanin pigmentation in coleoptile and pericarp	Yes, with TaPp-3	Reduces *TaPp-3* expression in pericarp. In complex with TaPp-3 activates its own expression. Probably shares redundant functions with TaPp-A1.	[56,253,258,259,269,275]
	*TaMpc1-A2* (*A genome MYB protein c1-1*)	5A	Probably, positive regulator of anthocyanin pigmentation in coleoptile and pericarp	Probably, with TaPp-3	Probably, is coregulator of *TaPp-1* genes, potentially activating the biosynthesis of uncolored anthocyanin precursors in association with TaPp-3	[56]
	*TaMpc1-D4/TaPL1-4D2* (*D genome MYB protein c1-4/Purple Plant 1-4*)	4D	Probably, positive regulator of anthocyanin pigmentation in coleoptile and pericarp	Probably, with TaPp-3 and TaMyc-A2	Probably, is coregulator of *TaPp-1* genes, potentially activating the biosynthesis of uncolored anthocyanin precursors in association with TaPp-3 or TaMyc-A2	[56,264]
	*TaMYB4D/TaMpc1*-*D3/TaPL1–4D3* (*D genome MYB protein c1*-*3*/*Purple Plant 1*-*3*)	4D	Potential positive regulator of anthocyanin pigmentation of aleurone	Possibly, with TaMY4D	Closely colocalizes with *TaMYC4D* and *TaF3′5′H* at the *Ba1* (*Blue aleurone 1*) locus	[75,255]

## 5. Conclusions

The accumulation of flavonoids is determined by a large number of tightly regulated enzymes of the phenylpropanoid biosynthetic pathway, providing temporal, spatial, and biochemical specificity of their accumulation. The regulation is primarily provided by three groups of proteins belonging to the MYC-like bHLH, R2R3-MYB, and WDR families, which form the MBW complexes and directly regulate the expression of structural genes. This process has been well studied in dicotyledonous plants, where intricate regulatory networks involve MYC and MYB transcription factors, alone or in complexes with WDR proteins, regulating the expression of each other and their coregulators.

In cereal plants, additional research of MBW proteins is needed due to their distinctive features that are of particular interest, including:

The presence of a large number of paralogs of the MYB and MYC transcription factors. Diversification of these paralogs has led to the spatial, temporal, and biochemical specialization, and their specific functions and characteristics are not yet fully understood. Interestingly, in many cases these genes are co-localized, forming multigenic clusters of similar or different regulatory genes and providing an additional layer of regulation. Additionally, multiple WDR regulators of flavonoid biosynthesis are potentially present in cereals, in contrast to dicotyledons, with a possibility of specialization. Further research is needed, as it provides opportunities to fine-tune flavonoid content in predetermined patterns.

Distinctive relationships between MYB, MYC, and WDR proteins. A competitive MBW complex formation, described in dicotyledons, is not known in cereals, while unique hierarchical regulatory systems, such as OsB2–OsC1–OsPAC1 in rice, are described. In some cases, such as the aerial vegetative parts of maize, the regulatory system formed by MYB and MYC transcription factors may function independently of WDR proteins. Overall, the regulatory networks formed by MBW proteins in cereals remain underexplored, and understanding their peculiarities is a very interesting aspect of cereal biology.

Unique regulatory systems involving MBW proteins. The unusual mechanism of regulation of MB or MBW complexes by EMSY-like proteins, such as RIF1 described in maize, remains unclear. In maize, RIF1-mediated formation of an alternative MB complex likely plays a key role in regulating early flavonoid biosynthesis genes. A similar function is performed by subgroup 7 R2R3-MYB transcription factors, unique to cereal plants, which have lost the ability to bind MYC proteins, leading to their specialization. In maize, this resulted in phlobaphene biosynthesis, which presents an intriguing research direction with the potential to finely tune the chemical composition of grain-based foods. Therefore, it is of particular interest to study the mechanisms of these alternative regulatory systems, as well as the associated differences in the activation of the expression of early and late structural genes of flavonoid biosynthesis.

In summary, the relationship of various MYC-like bHLH, R2R3-MYB, and WDR proteins, and the complexes they form, in plants is an excellent model for studying regulatory systems that determine the specificity of secondary metabolite accumulation. Despite the general similarity, the functioning of MBW proteins and their complexes in cereals has a number of unique features that remain incompletely understood.

## Figures and Tables

**Figure 1 ijms-26-00734-f001:**
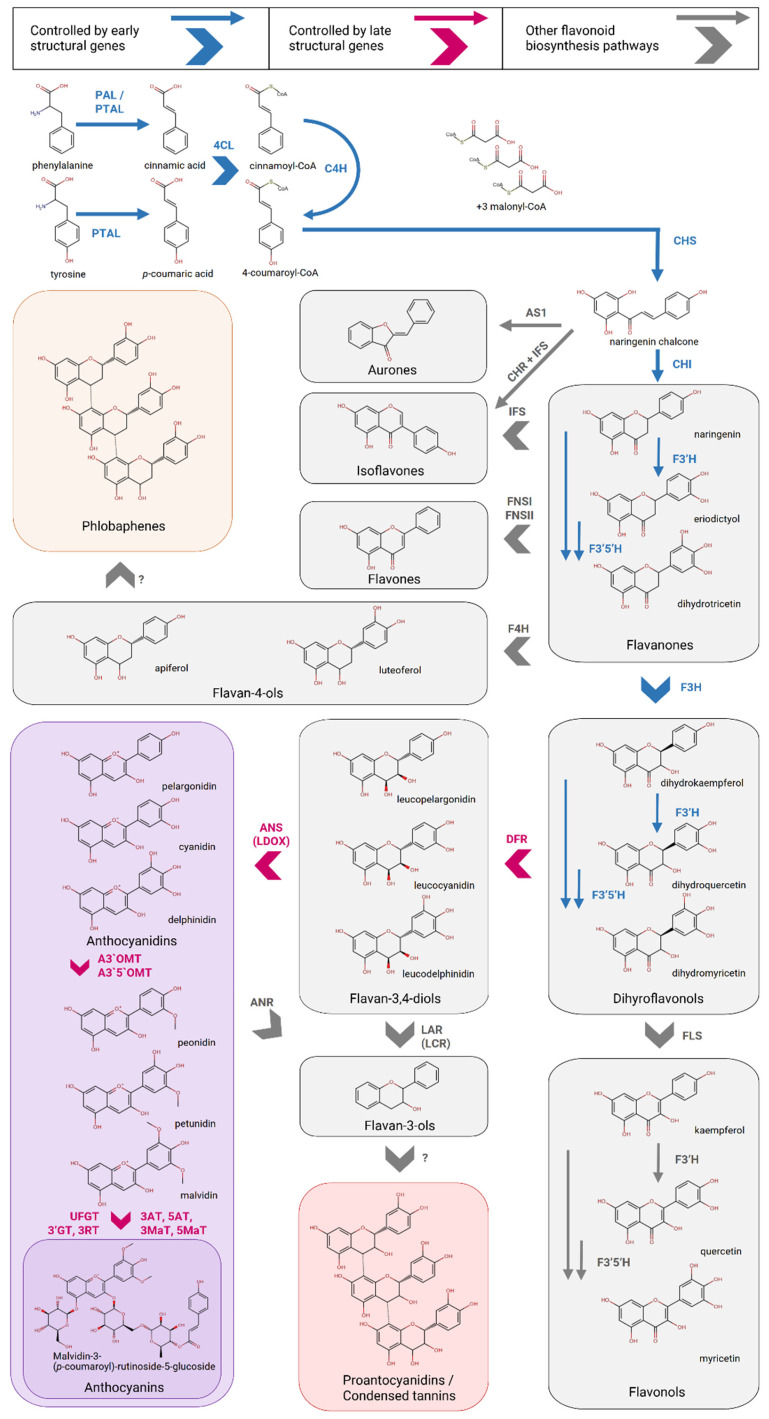
Schematic diagram of the biosynthetic pathways for anthocyanins and other flavonoid compounds. Blue and magenta arrows indicate the early and late steps of anthocyanin biosynthesis, respectively; and gray arrows indicate other flavonoid biosynthesis pathways. These reactions lead to the formation of diverse classes of flavonoids, highlighted in gray boxes, as well as flavonoid pigments, highlighted in colored boxes. The abbreviations near the arrows indicate the enzymes that catalyze the corresponding processes. PAL, phenylalanine ammonia lyase; PTAL, phenylalanine/tyrosine ammonia-lyase; 4CL hydroxycinnamate-CoA ligase; C4H, cinnamic acid 4-hydroxylase; CHS, chalcone synthase; AS1, aureusidin synthase; CHR, chalcone reductase; IFS, 2-hydroxyisoflavanone synthase; FNSI and II, flavone synthase I and II; F4H, flavanon 4-hydroxylase; CHI, chalcone isomerase; F3H, F3′H, and F3′5′H, flavone-3, -3′ and -3′5′-hydroxylase; FLS, flavonol synthase; DFR, dihydroflavonol-4-reductase; LAR/LCR, leucoanthocyanidin reductase/leucocyanidin reductase; ANS/LDOX, anthocyanidin synthase/leucoanthocyanidin dioxygenase; ANR, anthocyanidin reductase; A3′OMT and A3′5′OMT, anthocyanin 3′- and 3′5′-O-methyltransferase; UFGT, UDP-glucose: flavonoid-3-glycosyltransferase; 3′GT, anthocyanin 3′-O-beta-glucosyltransferase; 5GT, anthocyanin-5-O-glycosyltransferase; 3RT, UDP-rhamnose: anthocyanidin-3-glycoside rhamnosyltransferase; 3AT and 5AT, anthocyanin 3- and 5-aromatic acyltransferase, respectively; 3MaT, malonyl-CoA: anthocyanidin 3-O-glucoside-6″-O-malonyltransferase; 5MaT, malonyl-CoA: anthocyanidin 5-O-glucoside-6″-O-malonyltransferase.

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
