# Peer review of "Regulation of Flavonoid Biosynthesis by the MYB-bHLH-WDR (MBW) Complex in Plants and Its Specific Features in Cereals"

_ijms, 2025, doi:10.3390/ijms26020734_

Round 1

Reviewer 1 Report

Comments and Suggestions for Authors

The article "Regulation of flavonoid biosynthesis by the MYB-bHLH-WDR (MBW) complex in plants and its specific features in cereals" summarizes the regulatory mechanisms of flavonoid biosynthesis by the MYB-bHLH-WDR (MBW) complex. Additionally, it discusses the specificity of the MBW regulatory system in plants such as corn, rice, and wheat. I am very grateful to the authors for their focus on the regulatory mechanisms of MBW complexes in cereals, which is indeed a very interesting topic and of significant importance for the breeding. However, I still have some questions that I would like to discuss with the authors.

Firstly, the introduction part of the article mentions "This is one of the best-studied plant regulatory systems". Indeed, the MBW system has been the focus of many researchers for many years, and its mechanism is also relatively clear. I believe the innovative point of the article lies in the second half, that is, the specific regulatory mechanisms of the MBW system and its derived systems in different cereals. I suggest that you may consider reducing the discussion on the MBW mechanism and flavonoids in the first half, focusing more on the latter half.

Secondly, from Line 62 to 70, I have learned that the previous breeding goals for cereals were primarily aimed at reducing the content of flavonoid compounds within the grains, but it also mentions that there has been a growing number of attempts to develop cultivated varieties rich in flavonoids. This is puzzling, and perhaps a few sentences can explain the reasons behind this shift in breeding tendencies.

Furthermore, I have noticed that many of the referenced literature dates are quite outdated, and I suggest that it would be beneficial to include more up-to-date content regarding the MBW complex and its specific regulatory mechanisms across different species.

Lastly, there are some minor formatting issues. In the text, there are instances where italics is used incorrectly, such as in L932 where the Latin name of rice should be italicized, except for the genus and species names; the name of the person who named the species does not need to be italicized. There are also other misuses of italics in protein names and gene names that require careful proofreading. I also noticed that Figure 4 has a different font from the other images, and in Figure 5, apart from the gene names, the descriptive parts seem to be italicized as well, which is not recommended. It is suggested that the format of all images should be consistent.

Comments on the Quality of English Language

I suggest that there is still room for refinement in the use of some technical terms, such as "fertility of the plant" mentioned in L63, which does not seem to be a common usage in the field of botany.

Author Response

Dear Reviewer,

Thank you very much for taking the time to review this manuscript. Please find the detailed responses below and the corresponding revisions/corrections highlighted in the re-submitted files.

Comments 1: Firstly, the introduction part of the article mentions "This is one of the best-studied plant regulatory systems". Indeed, the MBW system has been the focus of many researchers for many years, and its mechanism is also relatively clear. I believe the innovative point of the article lies in the second half, that is, the specific regulatory mechanisms of the MBW system and its derived systems in different cereals. I suggest that you may consider reducing the discussion on the MBW mechanism and flavonoids in the first half, focusing more on the latter half.

Response 1: Thank you for pointing this out. We agree with this comment. Therefore, we broadened the second part of the introduction by emphasizing the differences between di- and monocotyledonous plants. But, because the review is addressed to a broad audience, we would rather keep the first part of the introduction. In our opinion, it is important to start with the common mechanism of MBW regulation of flavonoid biosynthesis.

Comment 2: Secondly, from Line 62 to 70, I have learned that the previous breeding goals for cereals were primarily aimed at reducing the content of flavonoid compounds within the grains, but it also mentions that there has been a growing number of attempts to develop cultivated varieties rich in flavonoids. This is puzzling, and perhaps a few sentences can explain the reasons behind this shift in breeding tendencies.

Response 2: Thank you, we have added the reason for these rather puzzling breeding tendencies: firstly the breeding goals were to maximize calorie content and produce whiter flour, which lead to decrease of flavonoids in grains. Studies of the last two decades showed possible beneficial effects of flavonoids on human health and have changed the breeding goals towards grains with higher flavonoid content.

Comment 3: Furthermore, I have noticed that many of the referenced literature dates are quite outdated, and I suggest that it would be beneficial to include more up-to-date content regarding the MBW complex and its specific regulatory mechanisms across different species.

Response 3: Thank you for your suggestions but please let us argue a little bit with that. The main idea of the review was to summarize as many relevant known facts about MBW regulation of flavonoid biosynthesis as possible to have the most complete picture of the problem. Many important researches were established more than thirty years ago, like studying R or B loci in maize, and we think it is better to reference available original articles regardless of their years of publication and then provide new additional details from the recent studies. In our opinion, it is more important to know original studies than their reinterpretation from more recent reviews. Moreover, 65 of 276 references are of the last five years (2019-2024) representing the recent state of the subject.

Comment 4: Lastly, there are some minor formatting issues. In the text, there are instances where italics is used incorrectly, such as in L932 where the Latin name of rice should be italicized, except for the genus and species names; the name of the person who named the species does not need to be italicized.

Response 4: Thank you very much for your thorough reading, we checked all the Latin names as well as persons who named the species and hope now it is all in order. Two instances of improperly italicized names of persons who named the species were corrected: L946 and L1034.

Comment 5: There are also other misuses of italics in protein names and gene names that require careful proofreading.

Response 5: We have proofread the manuscript for the correct use of italics in gene names L583, L595, L640, L641, L874, L1131, L1133, L1195 .

Comment 6: I also noticed that Figure 4 has a different font from the other images, and in Figure 5, apart from the gene names, the descriptive parts seem to be italicized as well, which is not recommended. It is suggested that the format of all images should be consistent.

Response 6: Thank you for bringing this to our attention. The font used for all illustrations is the same "Roboto" font from Google fonts package. The descriptive text is 10.6 points, locus names 11.5 points bold italics, gene names 7.7 points italics. As you suggested, we have changed italics to roman font in the descriptive parts of Figures 4 and 5 for better readability. Also, perhaps adding a picture into a text file may lead to loss of resolution. Please have a look at attached original images in .tif

Comment 7: Comments on the Quality of English Language. I suggest that there is still room for refinement in the use of some technical terms, such as "fertility of the plant" mentioned in L63, which does not seem to be a common usage in the field of botany.

Response 7: Thank you very much for the comment, we checked it, the term “plant fertility” is used in articles and reviews from the field of plant genetics of development and plant physiology:

https://doi.org/10.1016/j.phytochem.2004.07.028

https://www.sciencedirect.com/science/article/abs/pii/S0981942823005417

https://www.sciencedirect.com/science/article/pii/S1369526621001680

https://academic.oup.com/plcell/article/33/12/3604/6381586

Moreover, the term “restorers of fertility” is widely used for specific nuclear genes (Rf) having the ability to suppress the male-sterile phenotype. So, it is possible to conclude that “plant fertility” is a term used in a number of articles by plant geneticists. To resolve the issue of the term being not widely used we have rephrased the sentence “... since it strongly affects plant vitality and fertility."

Reviewer 2 Report

Comments and Suggestions for Authors

The manuscript presents a comprehensive review of the MBW regulatory complex in plants, detailing the structural and functional characteristics of its proteins, in addition to their mutual regulatory mechanisms. Furthermore, it engages in a discussion of the similarities and differences observed in dicotyledonous and monocotyledonous plants. With an emphasis on cereals, the manuscript systematizes rather scattered information about these regulators and their genes in cereal plants, as well as their distinctive features, intricacies of their functions and interactions with each other. While the manuscript offers valuable insights, its current length is deemed excessive, and there are some deficiencies in clarity and compactness. Therefore, it is recommended that the authors undertake a thorough revision of the paper to enhance its overall quality, necessitating major revisions.

Here are some major concerns

1、  Please avoid repetitive descriptions. Such as, “Flavonoids perform many functions necessary for the plant: serve as toxins and phytoalexins [1-3], protect the plant from an excessive light [4-7], function as pigments [8-10] or signaling molecules [11].” in line 37-40; “They serve as toxins and phytoalexins [1-3], protect the plant from an excessive light [4-7], function as pigments or signaling molecules [11].” in line 90-92.

2、  Please remove redundant expressions and the information that is not essential for grasping the central thesis. Such as, line 399-401, “The GL1 (GLABROUS1) gene determines trichome formation and encodes the R2R3-MYB transcription factor [170]. GL3 and EGL3 (ENHANCER OF GLABRA3) encode MYC TF and are responsible for the formation of epidermal structures.”; line 1093-1094, “Its expression is detected in many parts of the plant, including roots, stem, leaf sheaths, pericarp, lemma, and aleurone [255,263].”

3、  Please minimize the specific description of genes, and use the most concise expression to illustrate your point, especially in section 4.4.

4、  Please avoid elaborating the information that listed in the table again. For example, line 740-756.

5、  Line 537-541, “These data indicate that in Arabidopsis, competitive complex formation is a feature of the regulation of trichome formation and, apparently, does not play a significant role in the regulation of flavonoid biosynthesis (Figure 3C). However, this process plays a key role in the formation of trichomes (Figure 3B).” Please rewrite this sentence to simplify the expression.

6、  Line 37-40, why “serve as toxins” is a necessary function of flavonoids for plants?

7、  Line 62-71, it has been mentioned that “The accumulation of flavonoid compounds in cereals is a very important agricultural trait”, but why “Breeding of cereals such as maize, rice, barley, wheat and rye has been mainly aimed at reducing the content of flavonoid compounds in their grains, especially anthocyanins”? Please rephrase the statement to make the point is logical.

8、  Line 199-201, “The first experimental models”, but three species were mentioned? It may not be confusing if “first” is replaced by “initial”.

9、  In section “4.1.1. MYB transcription factors”, MYB transcription factors are commonly divided into four types, why only R1R2R3-type and R2R3-type have been mentioned? So far, there have been some reports on the regulation of flavonoids accumulation by 1R MYB. Have you paid any attention to whether they can participate in the formation of MBW complex? If not, what do you think are the possible reasons?

10、            The full gene name, such as C1 (Colorless 1), only needs to be marked at the first mention, and does not have to be repeated. Please check the full text.

11、            The font in Figure 1 is too small, especially for the groups in the structural formula.

Author Response

Dear Reviewer,

Thank you very much, we appreciate your thorough review of our manuscript. Below, we have provided detailed responses to your comments, with the corresponding revisions/corrections highlighted in the re-submitted files.

Comments 1:

Please avoid repetitive descriptions. Such as, “Flavonoids perform many functions necessary for the plant: serve as toxins and phytoalexins [1-3], protect the plant from an excessive light [4-7], function as pigments [8-10] or signaling molecules [11].” in line 37-40; “They serve as toxins and phytoalexins [1-3], protect the plant from excessive light [4-7], function as pigments or signaling molecules [11].” in line 90-92.

Response 1:

Thank you for bringing this to our attention. We have addressed this issue by removing the repetitive sentence from lines 90–92.

Comments 2:

Please remove redundant expressions and the information that is not essential for grasping the central thesis. Such as, line 399-401, “The GL1 (GLABROUS1) gene determines trichome formation and encodes the R2R3-MYB transcription factor [170]. GL3 and EGL3 (ENHANCER OF GLABRA3) encode MYC TF and are responsible for the formation of epidermal structures.”; line 1093-1094, “Its expression is detected in many parts of the plant, including roots, stem, leaf sheaths, pericarp, lemma, and aleurone [255,263].”

Response 2:

Let us disagree with this suggestion. This is an interesting observation that Arabidopsis genes, which are very similar to maize genes involved in anthocyanin biosynthesis regulation, in fact control very different processes, such as formation of epidermal structures. We believe that providing this information highlights an intriguing aspect of the functioning of MBW complexes and is relevant to the scope of the review.

Comments 3:

Please minimize the specific description of genes, and use the most concise expression to illustrate your point, especially in section 4.4.

Response 3:

Thank you for your suggestions, let us base our point of view. The main idea of the review is to summarize as many relevant known facts about regulation of flavonoid biosynthesis by MBW genes as possible to have the most complete picture of this topic. Making our own research we faced the problem of some inconsistency regarding MBW gene descriptions in some of the published articles. For example rice genes OsB1 is also referred to as Ra1, Pb, and Ps, while OsB2 is also designated Kala4 and S1. Unfortunately, there is still no consistent nomenclature for MBW genes in plants, particularly in cereals. Furthermore, summarizing the scattered information regarding the functions of these genes in cereals, particularly their tissue specificity, was a priority for us, as comprehensive summaries of such data are rare in existing publications. In many cases, providing detailed descriptions of gene functions and their unique features is essential to support subsequent statements. While we acknowledge that some gene descriptions might appear excessive, we believe they are necessary to present a complete and accurate overview of the subject.

Comments 4:

Please avoid elaborating the information that listed in the table again. For example, line 740-756.

Response 4:

Thank you for your comment. The brief descriptions of genes regulating flavonoid biosynthesis in various species are included in the table for the reader's convenience. However, the corresponding information in the text is an integral part of the narrative and supports the conclusions presented later. While the table serves primarily as a quick reference, the text provides context and deeper discussion. For these reasons, we respectfully disagree with your suggestion and believe that presenting this information in both formats is not redundant.

Comments 5:

Line 537-541, “These data indicate that in Arabidopsis, competitive complex formation is a feature of the regulation of trichome formation and, apparently, does not play a significant role in the regulation of flavonoid biosynthesis (Figure 3C). However, this process plays a key role in the formation of trichomes (Figure 3B).” Please rewrite this sentence to simplify the expression.

Response 5:

Thank you, we have rewritten the sentence as follows to simplify the expression: “Competitive complex formation in Arabidopsis seems to be the key feature of trichome formation regulation (Figure 3B), and, apparently, does not play a significant role in the regulation of flavonoid biosynthesis (Figure 3C).”

Comments 6:

Line 37-40, why “serve as toxins” is a necessary function of flavonoids for plants?

Response 6:

Thank you for your comment. This function is important for plants to defend against herbivores. To improve clarity, the sentence has been revised to: “Flavonoids perform many functions necessary for the plant: serve as phytoalexins, toxins against insects and herbivorous animals [1-3], protect the plant from excessive light [4-7], function as pigments [8-10] or signaling molecules [11].”

Comments 7:

Line 62-71, it has been mentioned that “The accumulation of flavonoid compounds in cereals is a very important agricultural trait”, but why “Breeding of cereals such as maize, rice, barley, wheat and rye has been mainly aimed at reducing the content of flavonoid compounds in their grains, especially anthocyanins”? Please rephrase the statement to make the point is logical.

Response 7:

Thank you, we have rephrased the statement to reflect the change in breeding goals: “Breeding of cereals such as maize, rice, barley, wheat and rye has been mainly aimed at reducing the content of flavonoid compounds in their grains, especially anthocyanins, to maximize calorie content and have white flour, therefore the majority of modern cultivated varieties are characterized by the absence of these secondary metabolites, or by their reduced amount, not only in the grain but also in the vegetative organs of the plant [38-42]. However, because of presumed beneficial effects on human health [32-34], attempts have been made to develop cultivated varieties enriched with flavonoid compounds.”

Comments 8:

Line 199-201, “The first experimental models”, but three species were mentioned? It may not be confusing if “first” is replaced by “initial”.

Response 8:

Thank you for pointing this out. We have corrected this mistake.

Comments 9:

In section “4.1.1. MYB transcription factors”, MYB transcription factors are commonly divided into four types, why only R1R2R3-type and R2R3-type have been mentioned? So far, there have been some reports on the regulation of flavonoids accumulation by 1R MYB. Have you paid any attention to whether they can participate in the formation of MBW complex? If not, what do you think are the possible reasons?

Response 9:

Thank you for your comment. Indeed, 1R-MYB proteins, also referred to as 3R-MYB proteins, play a significant role in regulating flavonoid biosynthesis. These proteins interact directly with MBW complexes, typically repressing their activity, which provides negative regulation and facilitates the formation of negative feedback loops. The role of 1R-MYB proteins in regulating flavonoid biosynthesis or related processes (such as trichome formation) through interactions with the MBW complex is discussed in section 4.3, particularly in the Arabidopsis and petunia regulatory systems. However, since only R2R3-MYB proteins are directly involved in the formation of MBW complexes activating the expression of structural genes, while R3-MYBs, along with other regulatory proteins, provide a secondary layer of flavonoid biosynthesis regulation (as mentioned in lines 618-634), their role was not emphasized in detail in this paper. As you have kindly pointed out, this was not explicitly mentioned in section 4.1.1. To improve clarity, the section title has been changed to 'R2R3-MYB Transcription Factors,' and the following statement has been added: “Diverse members of the R2R3-MYB family have been identified as key regulators of flavonoid biosynthesis, providing positive regulation of structural genes [18-22,27-28,47,64]. Proteins of the plant R3-MYB (1R-MYB) family, which have lost the R1- and R2-repeats, negatively regulate flavonoid biosynthesis or other processes regulated by MBW complexes by binding to bHLH proteins and interfering with R2R3-MYB transcription factors, thus providing additional regulation [20,27].”

Comments 10:

The full gene name, such as C1 (Colorless 1), only needs to be marked at the first mention, and does not have to be repeated. Please check the full text.

Response 10:

Thank you for pointing out this inaccuracy. We reviewed the entire text for excessive repetition of full gene names and corrected the identified instances. These corrections were made in L239, L299, L399, L328, L389, L391, L401, L402, L405, L742, L849, L850, L853, L876, L912, L952, L958, L963, L982, L1022, L1120, L1123, L1125, L1135, L1174, L1235. The full gene names in tables are retained on purpose for reader’s convenience to make tables autonomous from the main text and facilitate their reusability.

Comments 11:

The font in Figure 1 is too small, especially for the groups in the structural formula.

Response 11:

Thank you, we have redesigned the figure 1 to fit on page in a portrait orientation to ensure that the font used is at least 12pt in size. Also, perhaps adding a picture into a text file may lead to loss of resolution. Please have a look at attached original images in .tif

Reviewer 3 Report

Comments and Suggestions for Authors

The manuscript titled 'Regulation of flavonoid biosynthesis by the MYB-bHLH-WDR (MBW) complex in plants and its specific features in cereals' is well-crafted and clearly articulated. It can be aaccepted after addressing the minor points outlined below:

1. Please include details about the search process, such as the keywords used to identify relevant studies in the literature.

2. Clarify the inclusion and exclusion criteria applied during the search.

My primary concern, which has been highlighted by two inquiries, pertains to the methodology employed in the database search, specifically regarding the keywords utilized to identify pertinent articles. I kindly request that the authors provide a brief overview of the review methodology.

Author Response

Dear Reviewer,

Thank you for your review and appreciation of our work. Below, we have provided detailed responses to your comments regarding the methodology employed in the database search.

Comments 1:

Please include details about the search process, such as the keywords used to identify relevant studies in the literature.

Response 1:

Thank you for noting this aspect. The search for relevant literature was primarily conducted using the PubMed database (URL: https://pubmed.ncbi.nlm.nih.gov/), with additional searches performed in Google Scholar (URL: https://scholar.google.com/) and ScienceDirect (URL: https://www.sciencedirect.com/).

During the search process for literature relevant to the main scope of the article, primary keywords included "flavonoid" or specific subgroups of flavonoids (e.g., "anthocyanin" or "phlobaphene") along with family, tribe, genera, or species names in either English or Latin form (e.g., "cereals," "Triticeae," "maize," "Zea mays," "Arabidopsis," "Tibetan hulless barley") as well as clarifying keywords (e.g., "regulation," "MYC," "bHLH," "MYB," "WDR," "WD40," "MBW"). Typical search queries followed this format (using PubMed syntax): "(flavonoid) AND (Arabidopsis) AND (MBW)"; "(flavonoid) AND (maize) AND (MYC)"; "(anthocyanin) AND (Triticum aestivum) AND (bHLH)." In some cases, additional specificity was achieved by including terms such as "pericarp," "flower," "hierarchical regulation," "competitive complex formation," "light dependent," or by replacing gene family names (e.g., "MYC," "MYB," "WDR") with specific gene designations (e.g., "Purple pericarp," "Pp3," "ANT2," "Transparent testa 2," "TT8") to narrow the search field. Typically, multiple searches were performed using different combinations of synonyms, for example "(anthocyanin) AND (wheat) AND (bHLH)," "(anthocyanin) AND (Triticum aestivum) AND (bHLH)," and "(anthocyanin) AND (Triticum aestivum) AND (MYC)." In some cases, additional articles related to specific genes were identified through the "Bibliography" section of the gene’s NCBI Gene report (https://www.ncbi.nlm.nih.gov/gene/) or through the "Publications" section of the corresponding protein's UniProt entry (https://www.uniprot.org/).

Literature on the general features of the MYC, MYB, or WDR proteins was searched using queries such as "bHLH family," "(MYB family) AND (plants)," "(WDR family) AND (eukaryotes) AND (analysis)," or "(MYB family) AND (plants) AND (classification)."

Searches for articles related to structural genes and enzymes of flavonoid biosynthesis were performed using combinations of keywords such as "(flavonoid biosynthesis) AND (plants)," "(phlobaphene) AND (structural genes) AND (maize)," or "(flavonoid biosynthesis) AND (enzymes) AND (cereals)."

For articles focused on the functional roles of flavonoids in plants, searches combined keywords such as "flavonoid" or "anthocyanin" with terms like "function," "stress," "fertility," or "adaptivity." Searches focusing on the effects of flavonoids on human or animal health included keywords such as "flavonoid" or "anthocyanin" along with terms like "health," "food," "consumption," "human," "rat," or "cardiovascular."

Special attention was given to references cited in the reviewed literature, which often led to original sources and helped further refine the scope of the search, particularly for earlier articles.

Comments 2:

Clarify the inclusion and exclusion criteria applied during the search.

Response 2:

The articles included in the review were selected based on their relevance to the main scope of the manuscript and their contribution to the current understanding of the regulation of flavonoid biosynthesis in cereals and other plants by MBW complexes. The inclusion and exclusion criteria were as follows:

  • Inclusion Criteria:

1) For articles directly addressing MYC-like bHLH, R2R3-MYB, and WDR proteins and their genes regulating flavonoid biosynthesis original research articles were prioritized to ensure that the cited works were primary sources of discovery.

2) The main focus was on studies of maize, rice, barley, and wheat, as these are the most widely used and, consequently, studied cereal crops. For the general aspects of flavonoid biosynthesis regulation in plants, articles focused on Arabidopsis thaliana, petunia, snapdragon, and maize were prioritized, as these are the best-studied model systems in the field. Literature on other commonly studied species, primarily review articles, was also considered; however, only the most pertinent aspects were included in this paper.

3) For topics adjacent to the main scope of the review (e.g., flavonoid structural genes, general MBW protein features, or the functional roles of flavonoids), review articles were used to provide a comprehensive context. For specific statements, original research articles were incorporated when necessary.

4) Research providing novel insights or significantly broadening knowledge in the field was included. Preference was given to articles published in high-impact or well-regarded journals. Frequently cited articles were prioritized as indicators of foundational or widely recognized work. However, less-cited or niche articles were also incorporated if they presented unique or essential findings.

  • Exclusion Criteria:

1) Articles presenting redundant information already covered by other cited works were excluded.

2) Works lacking novel insights or providing generalized information without substantial evidence were excluded.

3) Articles that were secondary sources or reviews were excluded when primary sources of data could be cited instead.

Traditionally, the material and methods section in meta-analysis reviews include the inclusion and exclusion criteria for selecting the literature sources. Our review is not a meta-analysis, but, to clarify the point of concern, we have included them in the Introduction.

Reviewer 4 Report

Comments and Suggestions for Authors

This review addresses a challenging topic, summarizing the current state of knowledge on the regulation of flavonoid biosynthesis in cereal plants. The information provided by this review is of great importance as the accumulation of flavonoid compounds in cereals is a very important agricultural characteristic with a strong impact on plant vitality and fertility as well as food quality.

The introduction is robust, well considered, reflecting the current state of knowledge. The information included in this review is well-organized, presented in a well-structured manner.

The work has a significant contribution to the field providing a synthesis of relevant and current literature on the main aspects of the intriguing regulatory system of flavonoid biosynthesis in plants, along with valuable information on their distinctive features in cereals.

The study is fully reasoned and motivated as it provide an overview on flavonoid compounds and flavonoid pigments, flavonoid biosynthesis pathway as well as general mechanisms of regulation of flavonoid biosynthesis by MBW complexes.

The paper is scientifically sound and allows us to better understand the regulation of flavonoid biosynthesis and has good prospects to be a reference work, very useful for researchers focused on developing cereal varieties with the required flavonoid content and spatial distribution.

The references are relevant to the subject, covering a period of over 40 years, reflecting concern for the subject and a huge amount of work to render the information in a clear and well-organized form.

I recommend you to re-organize the Conclusions section in a more compact form. Please summarize here the main points addressed. I am aware of the complexity and comprehensiveness of the study, but it is important to provide some authentic findings based on your analysis.

Author Response

Dear Reviewer,

Thank you very much for your attentive review and thoughtful feedback on our work. Below, we provide our response to your comment regarding the re-structuring of the Conclusions section.

Comment:

I recommend you to re-organize the Conclusions section in a more compact form. Please summarize here the main points addressed. I am aware of the complexity and comprehensiveness of the study, but it is important to provide some authentic findings based on your analysis.

Response:

Following your suggestion, we have reorganized the Conclusions section in the re-submitted document to make it more concise. The main points addressed include the distinctive features of the MBW regulatory networks in cereals and their implications for future research and breeding:

  1. The presence of numerous specialized paralogs of MYB and MYC transcription factors, and potentially WDR proteins, whose diversity remains underexplored. Studying these factors provides opportunities to fine-tune flavonoid content in predetermined patterns.

  2. Unique relationships between MYB, MYC, and WDR proteins that are not yet fully understood.

  3. Regulatory systems involving MBW proteins, including those formed with EMSY-like proteins and subgroup 7 R2R3-MYB transcription factors, which specialize in regulating early stages of flavonoid biosynthesis. These systems present opportunities for breeding cereals with unique chemical compositions.

We hope these changes have made the Conclusions section more precise and aligned with your recommendations.

Round 2

Reviewer 1 Report

Comments and Suggestions for Authors

I have carefully reviewed the revised manuscript and the authors'response to the initial review comments.I am pleased to note that the authors have addressed the concerns raised in the first round of reviews in a satisfactory manner.The revisions made are appropriate,and the justifications provided for retaining certain aspects of the manuscript are reasonable.

The authors have demonstrated a clear understanding of the feedback and have supplemented some content,which has enhanced the overall quality and coherence of the paper,making it easier for readers to understand.Considering the target audience of the article,I agree with the authors'decision to retain the first part of the introduction.

Given the improvements and the authors'thorough response to the initial comments,I recommend that the manuscript be accepted for publication.It represents a valuable contribution to the field and will be of interest to the readers of the journal.

Reviewer 2 Report

Comments and Suggestions for Authors

I would like to thank the authors' comprehensive responses to my comments and the revisions they have implemented in the manuscript, and now I have no further questions.